# Towards Interpretability Without Sacrifice: Faithful Dense Layer Decomposition with Mixture of Decoders

**James Oldfield**[m,q*]   **Shawn Im**[m]   **Sharon Li**[m]   **Mihalis A. Nicolaou**[c,i]
**Ioannis Patras**[q]   **Grigorios G Chrysos**[m]

[m] University of Wisconsin–Madison   [q] Queen Mary University of London   [c] University of Cyprus
[i] The Cyprus Institute

## Abstract

Multilayer perceptrons (MLPs) are an integral part of large language models, yet their dense representations render them difficult to understand, edit, and steer. Recent methods learn interpretable approximations via neuron-level sparsity, yet fail to faithfully reconstruct the original mapping–significantly increasing model's next-token cross-entropy loss. In this paper, we advocate for moving to *layer*-level sparsity to overcome the accuracy trade-off in sparse layer approximation. Under this paradigm, we introduce Mixture of Decoders (MxDs). MxDs generalize MLPs and Gated Linear Units, expanding pre-trained dense layers into tens of thousands of specialized sublayers. Through a flexible form of tensor factorization, each sparsely activating MxD sublayer implements a linear transformation with full-rank weights–preserving the original decoders' expressive capacity even under heavy sparsity. Experimentally, we show that MxDs significantly outperform state-of-the-art methods (e.g., Transcoders) on the sparsity-accuracy frontier in language models with up to 3B parameters. Further evaluations on sparse probing and feature steering demonstrate that MxDs learn similarly specialized features of natural language–opening up a promising new avenue for designing interpretable yet faithful decompositions. Our code is included at: https://github.com/james-oldfield/MxD/.

## 1   Introduction

One strategy for addressing concerns about large language models' (LLMs) [1, 2, 3] behavior is via a bottom-up approach to understanding and controlling the network internals–developing models of how and where human-interpretable features are represented in LLMs and how they affect the output [4, 5, 6]. Such a mechanistic understanding has proved helpful for a number of issues relating to safety and transparency, from controlling refusal of harmful requests [7] to detecting generation of unsafe code [6] and latent model knowledge [8].

However, developing models of LLMs' internals faces challenges due to the dense nature of their representations [9, 10]. Indeed, many studies have found that individual neurons in MLP layers encode *multiple* distinct concepts. Rather than human-interpretable features being neatly aligned with individual neurons, they are often distributed across many [11, 12]. As a result, it is not straightforward to cleanly isolate specific concepts of interest in the models' latent token representations.

Traditionally, imposing constraints on model form has offered a way to instill more predictable properties or structure. Indeed, there is a rich history of success with constraints in machine learning: from parts-based representations through non-negativity [13, 14], to structure through low-rankness or assumptions on geometry [15, 16]. With the particular issues posed by dense representations

---

*Corresponding author: jamesalexanderoldfield@gmail.com. Work done whilst at UW-Madison.

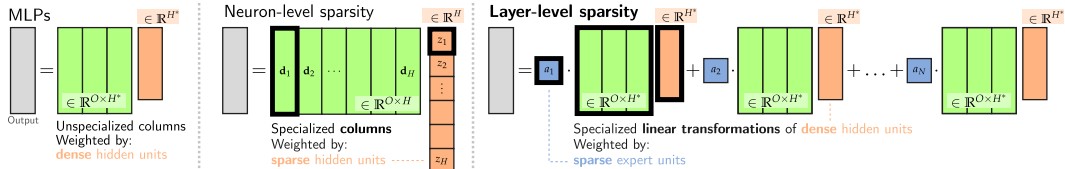

Figure 1: **Units of specialization for sparse layer variants**: *Neuron*-level sparsity of existing sparse MLPs [27, 26] (center) vs *layer*-level sparsity (right), which the proposed Mixture of Decoders (MxD) layer enables at scale. For GPT2, the dimensions are: $O = 768$, $H^* = O \cdot 4$, $H \approx N \approx O \cdot 32$.

in LLMs, *specialization through sparsity* has re-emerged as a dominating strategy for learning more interpretable representations. With prior work showing that sparser models both aid human explanation [17] and achieve higher scores on LLM-based auto-interpretability metrics [18, 19], sparsity is often used as a proxy for interpretability [20, 21]. To this end, many recent works– such as sparse autoencoders [22, 23, 6]–take inspiration from traditional sparse dictionary learning methodologies [24, 25], re-writing pre-trained LLMs' activations as sparse, non-negative linear combinations of atoms in a learned overcomplete basis. However, as argued in [26], such approaches do not learn the functional mechanisms of LLMs' layers, and their inherent post-hoc nature demands additional parameters and computation on top of the base models.

One alternative approach is to directly replace layers with more interpretable equivalents [28], such as with wide MLPs with sparsity constraints. Transcoders [27, 29, 30, 26] (TCs) are a recent example of this, training new MLPs to mimic the functional behavior of MLPs with *sparse* hidden units, which have recently been shown to also learn more interpretable features [26]. Thus, instead of relying on external post-hoc analysis, sparse MLP layers offer a way to distill specialized features directly into the model's forward pass itself.

Both of the above methods for learning specialized features fall into the same category of what one may call 'neuron-level sparsity'. Dictionary learning methods restrict the number of non-zero elements used from a learned dictionary, whilst sparse MLPs [27] limit the number of active rows used from a learned 'decoder' matrix. At its core, whilst this constraint is useful for interpretability, it is too restrictive–often heavily trading off accuracy for sparsity, poorly reconstructing the original model components [31, 28]. We argue that preserving the base models' performance is a crucial component of sparse MLP layer approximations for the following two key reasons:

1. **Model faithfulness**: sparse layers that poorly approximate the original layers risk missing critical intricacies of the base models' behavior or latent features [32]. Conversely, an accurate reconstruction (yielding similar downstream next-token loss) is some evidence that the combination of newly learned subcomputations faithfully emulates the base model.

2. **Practical adoption**: sparse layers that closely preserve base models' performance are capable of *replacing* the existing MLPs, directly integrating specialized computation into the native forward pass. Otherwise, downstream use of the sparse layers' features must run on top of the base models' computation. This introduces additional inference-time cost to every forward pass, and restricts any analysis to post-hoc settings.

In this paper, we advocate for moving from *neuron*-level to *layer*-level sparsity (as illustrated in Figure 1) to address this. We propose the *Mixture of Decoders (MxD)* layer to overcome the sparsity-accuracy trade-off through scalable, resource-efficient conditional computation. Rather than individual vectors, MxDs learn interpretable sublayers as atomic units of specialization. This faithfully mirrors the functional form of dense layer we wish to approximate, and allows MxDs to readily generalize to modern MLP variants (i.e., the Gated Linear Unit [33]).

At a technical level, MxDs are constructed via a flexible tensor factorization [34] with the Hadamard product [35]. Through their parameter efficiency, MxDs scale the number of specialized layers far beyond what is feasible with classic sparse mixture of experts (MoEs) [36], and recover prior adapter-based MoEs [37, 38] as a special case. Crucially, we prove that the proposed tensor factorization in MxDs leads to each 'expert' sublayer implementing a linear transformation with full-rank weights–allowing faithful reconstruction even under heavy sparsity. Empirically, we demonstrate that MxDs significantly outperform alternative sparse MLP layers such as Transcoders [27] and Skip Transcoders

[26] on the sparsity-accuracy frontier. In addition to their faithfulness, MxDs remain competitive with the SOTA on interpretability metrics. **Our contributions can be summarized as follows:**

- We propose *Mixture of Decoders*, an instance of a flexible class of parameter-efficient MoE through Hadamard product-factorized weight tensors.
- We prove that each specialized MxD expert's weights inherit up to the same rank as the original MLP's decoder, providing faithful approximation even in very sparse models.
- Across 108 sparse layers in 4 LLMs (with up to 3B parameters) MxDs (i) pareto-dominate existing techniques on the sparsity-accuracy frontier yet (ii) remain competitive on 34 sparse probing and steering tasks, validating the interpretability of the learned experts.

## 2 Methodology

We first recall the technical details of language models' MLP layers and existing approaches to sparse approximations in Section 2.1. We then introduce the proposed MxD in Section 2.2, outlining the attractive rank properties it inherits in Section 2.3 and factorized implementation in Section 2.4. We conclude with extensions to modern MLP layers in Section 2.5.

### 2.1 Preliminaries

Let $\mathbf{x} \in \mathbb{R}^I$ be the pre-MLP latent representation of a specific token at a given layer. Omitting bias terms throughout for brevity, the GPT2-style MLP layer produces the output vector $\mathbf{y} \in \mathbb{R}^O$ as:

$$\mathrm{MLP}(\mathbf{x}) = \mathbf{D}^{*\top}\mathbf{z}^* \in \mathbb{R}^O, \quad \text{with } \mathbf{z}^* := \phi\big(\mathbf{E}^{*\top}\mathbf{x}\big) \in \mathbb{R}^{H^*}, \tag{1}$$

where $\mathbf{E}^* \in \mathbb{R}^{I \times H^*}$, $\mathbf{D}^* \in \mathbb{R}^{H^* \times O}$ are the learnable 'encoder' and 'decoder' parameters respectively, and $\phi(.)$ is an activation function, often a GELU [39]. We use $^*$ to denote the weights/dimensions of the pre-trained base LLM.

**Sparse approximations** One approach to learning interpretable features in MLPs is to train new, wider MLPs with *sparse* hidden units to reconstruct the original layer's outputs [27, 26, 30, 29], reminiscent of dictionary learning techniques [25]. In general, sparse MLPs share the model form:

$$\mathrm{SMLP}(\mathbf{x}) = \mathbf{D}^\top\mathbf{z} = \sum_{h=1}^{H} z_h \mathbf{d}_h \in \mathbb{R}^O, \quad \text{with } \mathbf{z} := \mathcal{S}\big(\mathbf{E}^\top\mathbf{x}\big) \in \mathbb{R}^H, \tag{2}$$

where $\mathcal{S}(.)$ is a sparsity-inducing function (such as the top-$K$ [23] activation used in this paper). Here, the dimensionality of sparse MLPs' learnable weights $\mathbf{E} \in \mathbb{R}^{I \times H}$, $\mathbf{D} \in \mathbb{R}^{H \times O}$ are set as $H \gg H^*$ such that the hidden layer is significantly larger than that of the original MLP. The original post-MLP output vectors are approximated as a $K$-sparse, non-negative linear combination of the rows $\mathbf{d}_n$ of a newly learned decoder matrix. Whilst this model form has been shown to learn interpretable, specialized features $z_h$ in language models [27, 26], their poor reconstruction is of questionable faithfulness and limits their use as a layer replacement in practice.

### 2.2 Mixture of Decoders

We now detail the proposed Mixture of Decoders (MxD) layer, which overcomes the sparsity-accuracy trade-off by treating sparsely activating linear layers as the atomic unit of specialization. We approximate the original MLP with a conditional combination of $N$ *linear transformations*:

$$\mathrm{MxD}(\mathbf{x}) = \sum_{n=1}^{N} a_n(\mathbf{W}_n^\top\mathbf{z}) \in \mathbb{R}^O, \tag{3}$$

where $\mathbf{a} := \mathcal{S}\big(\mathbf{G}^\top\mathbf{x}\big) \in \mathbb{R}^N$ are *sparse* 'expert coefficients' from learnable gating matrix $\mathbf{G} \in \mathbb{R}^{I \times N}$, and $\mathbf{z} := \phi\big(\mathbf{E}^\top\mathbf{x}\big) \in \mathbb{R}^H$ is the *dense* output from an encoder. Here, $\boldsymbol{\mathcal{W}} \in \mathbb{R}^{N \times H \times O}$ is a third-order tensor of parameters collating all $N$ experts' decoder weights $\boldsymbol{\mathcal{W}}(n,:,:) = \mathbf{W}_n \in \mathbb{R}^{H \times O}$. In MxDs, we use a large $N$ to scale the feature specialization, and set $H := H^*$ to match the original MLP's smaller hidden dimension.

With the gate routing each token to just its top-$K$ experts, each $\mathbf{W}_n \in \mathbb{R}^{H \times O}$ receives a gradient signal from only a specific set of semantically similar tokens. This implicit clustering naturally leads experts to specialize in feature-specific subcomputations, while collectively covering the layer's full functionality. MxDs in Equation (3) also directly inherit the MLP layers' original functional form, avoiding the need to impose sparsity and non-negativity constraints on the hidden units $\mathbf{z} \in \mathbb{R}^H$. However, MxD decoders naively require a prohibitive $NHO$ parameters–preventing $N$ from scaling to tens of thousands of specialized components. To achieve parameter-efficiency whilst retaining layer capacity for faithful layer approximation, we parameterize MxDs' third-order weight tensor $\mathcal{W} \in \mathbb{R}^{N \times H \times O}$ specifically to yield full-rank expert weights, defined elementwise as:

$$\mathcal{W}(n, h, :) = \mathbf{c}_n * \mathbf{d}_h \in \mathbb{R}^O, \qquad \forall n \in \{1, \ldots, N\},\ h \in \{1, \ldots, H\}, \tag{4}$$

where $*$ is the *Hadamard product* [34, 35], and $\mathbf{c}_n$, $\mathbf{d}_h \in \mathbb{R}^O$ are the rows of learnable weights $\mathbf{C} \in \mathbb{R}^{N \times O}$, $\mathbf{D} \in \mathbb{R}^{H \times O}$. Intuitively, $\mathbf{D}$ implements a base transformation modulated by the $N$ specialized units in $\mathbf{C}$. Additional technical motivation for this parameterization with tensor methods can be found in Appendix A.3. This brings MxDs' parameter count down significantly to $O \cdot (N + H)$ from $NHO$ in Equation (3) with $N$ full decoders. One can then vary $N$ to parameter-match sparse MLP layers. We next detail how this design (i) retains expressivity in each unit for faithful layer approximation under sparsity in Section 2.3 and (ii) yields a simple forward pass in Section 2.4.

## 2.3 MxDs are rank-preserving

In the original LLM, the linear transformation from the hidden units to the output is constrained by the rank of the original MLP's decoder matrix $\mathbf{D}^* \in \mathbb{R}^{H^* \times O}$. Under only mild technical conditions, *every* expert's weight matrix in MxDs inherits the rank of $\mathbf{D} \in \mathbb{R}^{H \times O}$, thus allowing it to match that of the original MLP's decoder, despite its parameter-efficiency:

**Lemma 1** (Decoder rank preservation). *We can materialize linear expert $n$'s weight matrix as* $\mathcal{W}(n, :, :) = \mathbf{W}_n = \mathbf{D} \operatorname{diag}(\mathbf{c}_n) \in \mathbb{R}^{H \times O}$. *Assuming* $\operatorname{diag}(\mathbf{c}_n) \in \mathbb{R}^{O \times O}$ *is a diagonal matrix with no zeros along its diagonal (and thus invertible), we then have*

$$\operatorname{rank}(\mathbf{W}_n) = \operatorname{rank}(\mathbf{D} \operatorname{diag}(\mathbf{c}_n)) = \operatorname{rank}(\mathbf{D}).$$

The proof is found in Appendix A.1, which first derives the matrix-valued expression for each expert from Equation (4) and then applies a standard rank equality. At a sparsity level of $K$, each MxD output vector is a weighted sum of $K$-many linear transformations (each with potentially full-rank weights) of the dense hidden units $\mathbf{z}$. As a result, MxDs retain layer capacity even under high sparsity. Sparse MLPs' hidden units have only $K$ non-zero elements in contrast–each output in Equation (2) is therefore confined to a $K$-dimensional subspace of $\mathbb{R}^O$, potentially limiting the capacity of sparse MLPs to faithfully approximate the original mapping in the small $K$ regime desirable for interpretability (mirroring speculations by [26]). Further, whilst alternative soft linear MoEs achieve scalability through low-rankness [40], Lemma 1 states that no such rank constraints are present in MxDs. For approximating existing MLP layers where low-rank assumptions may not hold, MxDs are consequently a more suitable class of conditional layer.

## 2.4 Factorized forward pass

MxDs compute a linear combination of $N$ linear transformations of the dense vector. With the proposed Hadamard-factorized weights, this yields a simple implementation.

**Lemma 2** (Hadamard-factorized MoE forward pass). *Let $\mathbf{z} \in \mathbb{R}^H$ and $\mathbf{a} \in \mathbb{R}^N$ denote the MLP hidden units and expert coefficients respectively. Further, denote the decoder matrices as $\mathbf{C} \in \mathbb{R}^{N \times O}$, $\mathbf{D} \in \mathbb{R}^{H \times O}$ parameterizing $\mathcal{W} \in \mathbb{R}^{N \times H \times O}$. MxD's forward pass can be re-written as:*

$$\mathrm{MxD}(\mathbf{x}) = \sum_{n=1}^{N} a_n (\mathbf{W}_n^\top \mathbf{z}) = (\mathbf{C}^\top \mathbf{a}) * (\mathbf{D}^\top \mathbf{z}). \tag{5}$$

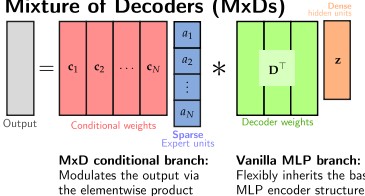

**Mixture of Decoders (MxDs)**

**MxD conditional branch:** Modulates the output via the elementwise product

**Vanilla MLP branch:** Flexibly inherits the base MLP encoder structure

Figure 2: **Mixture of Decoders** extends the base MLP/GLU layers with a conditional 'expert' branch, modulating the MLP's outputs.

The proof is found in Appendix A.2. We include a notebook at https://github.com/james-oldfield/MxD/blob/main/form-equivalence.ipynb showing the equivalence in PyTorch. Further, please see Appendix A.5 for a discussion of how the Hadamard factorization relates

Table 1: **Model formulations of related work**: $\mathbf{x} \in \mathbb{R}^I$, $\mathbf{y} \in \mathbb{R}^O$ are the pre- and post-MLP representations respectively, $\mathbf{z}$ are the hidden units, and $\mathbf{a}$ is the vector of the 'expert coefficients' for MxD. Model-specific encoders/decoders $\mathbf{E}$, $\mathbf{D}$ map between the hidden units and output.

| | MLPs [3] | SAEs [22] | Transcoders [27] | Skip Transcoders [26] | **MxDs (Ours)** |
|---|---|---|---|---|---|
| Model form | $\mathbf{y} = \mathbf{D}^{*\top}\mathbf{z}^*$ | $\mathbf{y} \approx \mathbf{D}^\top\mathbf{z}$ | $\mathbf{y} \approx \mathbf{D}^\top\mathbf{z}$ | $\mathbf{y} \approx \mathbf{D}^\top\mathbf{z} + \mathbf{S}^\top\mathbf{x}$ | $\mathbf{y} \approx \sum_n a_n \left(\mathbf{W}_n^\top \mathbf{z}\right)$ |
| Sparse component | None | $\mathbf{z} = \mathcal{S}\left(\mathbf{E}^\top\mathbf{y}\right) \in \mathbb{R}^H$ | $\mathbf{z} = \mathcal{S}\left(\mathbf{E}^\top\mathbf{x}\right) \in \mathbb{R}^H$ | $\mathbf{z} = \mathcal{S}\left(\mathbf{E}^\top\mathbf{x}\right) \in \mathbb{R}^H$ | $\mathbf{a} = \mathcal{S}\left(\mathbf{G}^\top\mathbf{x}\right) \in \mathbb{R}^N$ |

to prior parameter-efficient MoEs with element-wise scaling [37], and Appendix B.6 for performance/computational cost comparisons.

## 2.5 Extending MxDs to GLUs

In contrast to methods imposing neuron-level sparsity [22, 27, 26], MxDs do not make assumptions about the base layer's encoder architecture or activation function. As a result, MxDs readily generalize to alternative architectures such as the Gated Linear Units (GLUs) [33] used in recent LLMs [1, 2]. Recall that GLUs' hidden units are computed as $\mathbf{z}_{\text{GLU}} = \psi(\mathbf{E}_{\text{GLU}}^\top\mathbf{x}) * \left(\mathbf{E}^\top\mathbf{x}\right) \in \mathbb{R}^H$, with additional GLU parameters $\mathbf{E}_{\text{GLU}} \in \mathbb{R}^{I \times H}$ and GLU activation function $\psi$ (e.g., Swish [1]). By substituting in the GLU hidden representations, MxDs straightforwardly extend the GLU model form too:

$$\text{MxD}_{\text{GLU}}(\mathbf{x}) = \sum_{n=1}^N a_n \, \mathbf{W}_n^\top \left( \underbrace{\psi(\mathbf{E}_{\text{GLU}}^\top\mathbf{x}) * \left(\mathbf{E}^\top\mathbf{x}\right)}_{\text{GLU hidden units}} \right) = \left(\mathbf{C}^\top\mathbf{a}\right) \, * \, \mathbf{D}^\top\left(\psi(\mathbf{E}_{\text{GLU}}^\top\mathbf{x}) * \left(\mathbf{E}^\top\mathbf{x}\right)\right)$$

where $\mathbf{a} := \mathcal{S}\left(\mathbf{G}^\top\mathbf{x}\right) \in \mathbb{R}^N$ are the expert units, and $\mathbf{W}_n = \mathbf{D}\operatorname{diag}(\mathbf{c}_n) \in \mathbb{R}^{H \times O}$ as before. For a technical discussion of GLUs and their relationship to MxDs, we refer readers to Appendix A.4– through the theoretical results developed in this paper, we show that GLU encoders themselves can be viewed as a mixture of rank-1 linear experts (in contrast to the rank-preserving MxDs).

## 3 Experiments

The experimental section in the main paper is split into two parts. Section 3.1 first demonstrates how MxDs perform significantly better on the accuracy-sparsity frontier as sparse MLP layer approximations on 4 LLMs. We then demonstrate in Section 3.2 that MxD's features retain the same levels of specialization through sparse probing and steering evaluations. Thorough ablation studies, experiments with matrix rank, and comparisons to low rank MoEs are presented in Appendix B.

## 3.1 Sparse approximations of MLPs in LLMs

In this section, we perform experiments approximating LLMs' existing feed-forward layers with sparse MLPs, establishing that MxDs better navigate the sparsity-accuracy frontier, more faithfully approximating the base models' MLPs than the SOTA baseline methods.

**Implementation details** We train on 4 base models: GPT2-124M [3], Pythia-410m, Pythia-1.4b [41], and Llama-3.2-3B [1] with up to 80k experts/features. We train all sparse layers on a total of 480M tokens of OpenWebText [42], with learning rate $1e-4$ and a context length of 128, initializing the output bias as the empirical mean of the training tokens, and $\mathbf{D}$ in MxDs as the zero-matrix (following [26]). We vary $N$ in MxD layers to parameter-match Transcoders in all experiments, with parameter counts and dimensions shown in Table 2. For Llama3.2-3B, we use the Swish-GLU variant of MxD and GELU-MLP MxDs for the other three models, matching the architectures of their base encoders. Through ablation studies in Appendix B.8 we show that MxDs using the GELU/GLU variants are much more accurate layer approximators than the ReLU variants. Full experimental details are included in Appendix D. Whilst we do not have the computational resources to show similarly thorough experiments on even larger models, we expect MxDs to scale just as well as sparse MLPs to models with tens of billions of parameters or more.

Table 2: Sparse layer parameters/dimensions: $H$ denotes the size of the layers' hidden units and $N$ is the expert count. MxDs perform almost as many linear transformations as the baselines have features.

| | GPT2-124M | | | Pythia-410M | | | Pythia-1.4B | | | Llama-3.2-3B | | |
|---|---|---|---|---|---|---|---|---|---|---|---|---|
| Model | Params | $H$ | $N$ | Params | $H$ | $N$ | Params | $H$ | $N$ | Params | $H$ | $N$ |
| Transcoders [27] | 37.7M | 24,576 | — | 67.1M | 32,768 | — | 268.5M | 65,536 | — | 604M | 98,304 | — |
| Skip Transcoders [26] | 38.4M | 24,576 | — | 68.2M | 32,768 | — | 272.7M | 65,536 | — | 614M | 98,304 | — |
| **MxDs** | 37.7M | 3072 | 21,490 | 67.1M | 4096 | 28,658 | 268.4M | 8192 | 57,330 | 604M | 8202 | 86,015 |

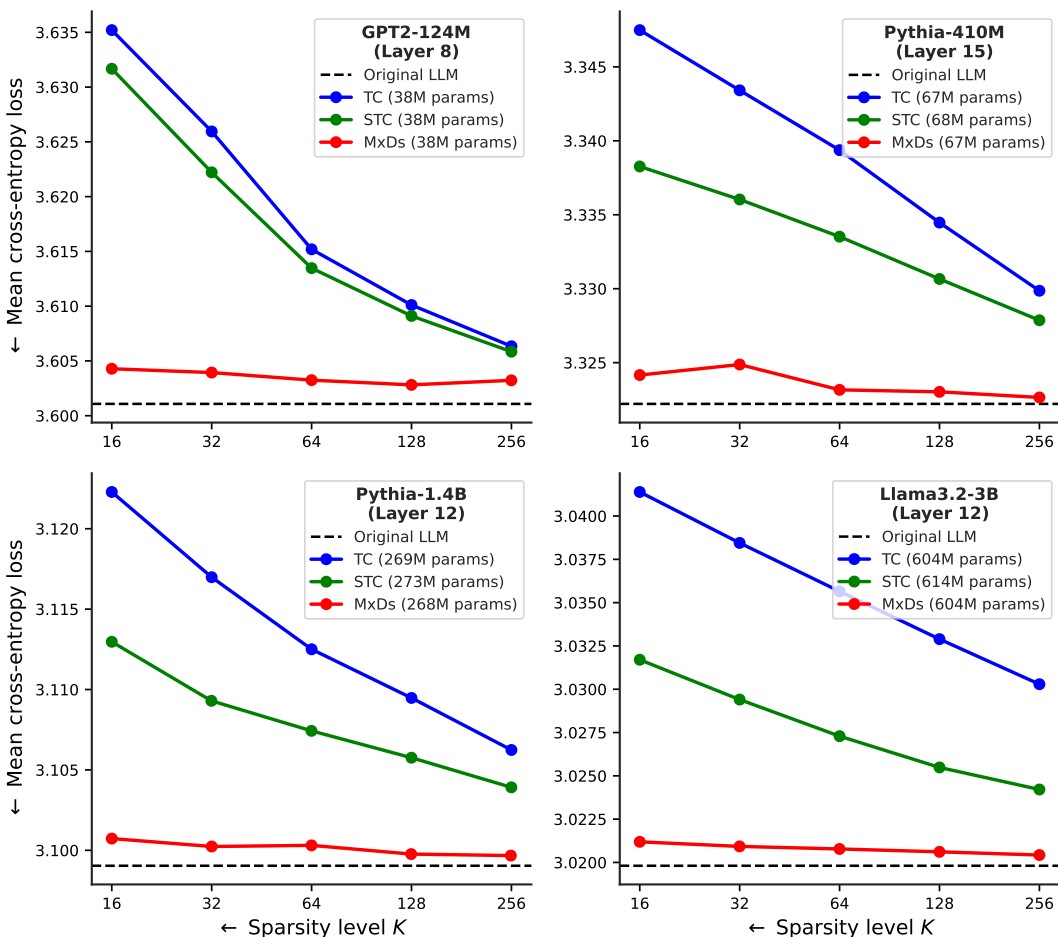

Figure 3: Model cross-entropy loss preserved when replacing MLPs with Transcoders [27], Skip Transcoders [26], and MxDs, as a function of the number of active units $K$ (hidden neurons/experts). We highlight that MxDs have consistently lower loss at all levels of sparsity.

**Objective function** Given the frozen weights of the MLP, we train sparse layers to minimize the normalized reconstruction loss between its output and that of the original MLP layer with objectives of the form $\mathcal{L} = \mathbb{E}_{\mathbf{x}} \left[ \frac{||\mathrm{MLP}(\mathbf{x}) - f(\mathbf{x})||_2^2}{||\mathrm{MLP}(\mathbf{x})||_2} \right]$, where $f(.)$ denotes the various learnable sparse MLP layers. This follows the protocol of past work [27, 26], where the new sparse layers' parameters alone are trained directly on the output of the MLP. To compare with recent work [26], we adopt the TopK activation function [23] for sparsity-inducing function $\mathcal{S}(.)$, removing the need for an additional sparsity penalty. Please see Appendix A.6 for details on the TopK activation function.

### 3.1.1 Results: sparsity vs faithfulness

We train an exhaustive set of 60 sparse MLP approximations across 4 diverse LLMs with up to 3B parameters. We show in Figure 3 the resulting downstream base model cross-entropy loss when using the trained sparse layers in place of the original MLPs. As can be seen, not only do the proposed

MxD layers outperform Transcoders [27] notably, but **model performance is similarly preserved at all sparsity levels in MxD layers**. Please also see Figure 9 for results with normalized MSE, where MxDs' reconstruction errors are up to an order of magnitude smaller. Full results on additional layers are included in Appendix B.3 for 48 more trained sparse layers.

The recent 'Skip Transcoders' (STCs) [26], introduce an additional $IO$ parameters with a skip connection $\mathbf{S} \in \mathbb{R}^{I \times O}$ mapping the input directly to the output with $\mathbf{y} \approx \mathbf{D}^\top \mathbf{z} + \mathbf{S}^\top \mathbf{x}$. STC layers thus have considerably more parameters (e.g., STCs on `llama3.2-3B` have 10M more parameters than MxDs). Despite the smaller parameter counts, we find MxDs consistently outperform STCs on the sparsity-accuracy frontier, attesting to the benefits of MxDs' model form.

### 3.1.2 Faithfulness in output space

Next, we perform experiments comparing the faithfulness of sparse layers as their computation propagates to the model output space. We sample 16 future tokens with the base model and then measure how similar the same generations are when the target MLP layer is replaced with the sparse layers. We use 512 text snippets from OpenWebText, and take the first 4 words of each as the initial prompts, generating 16 future tokens after each prompt. We plot in Figures 4a and 4b the percentage of the samples' continuations that are identical in the original LLM and hooked LLMs up to $n$ future tokens ahead. We note that this is a rather punishing task–any small deviations quickly compound as $n$ grows. Despite this, we see that the MxDs match the future token generations far better than the baselines, exhibiting more faithfulness in model output space (as well as in latent space).

Please see qualitative examples of the first 8 prompts and the subsequent 'diffs' (using Python 3's `difflib`) of the generated tokens in the Appendix in Figures 7 and 8.

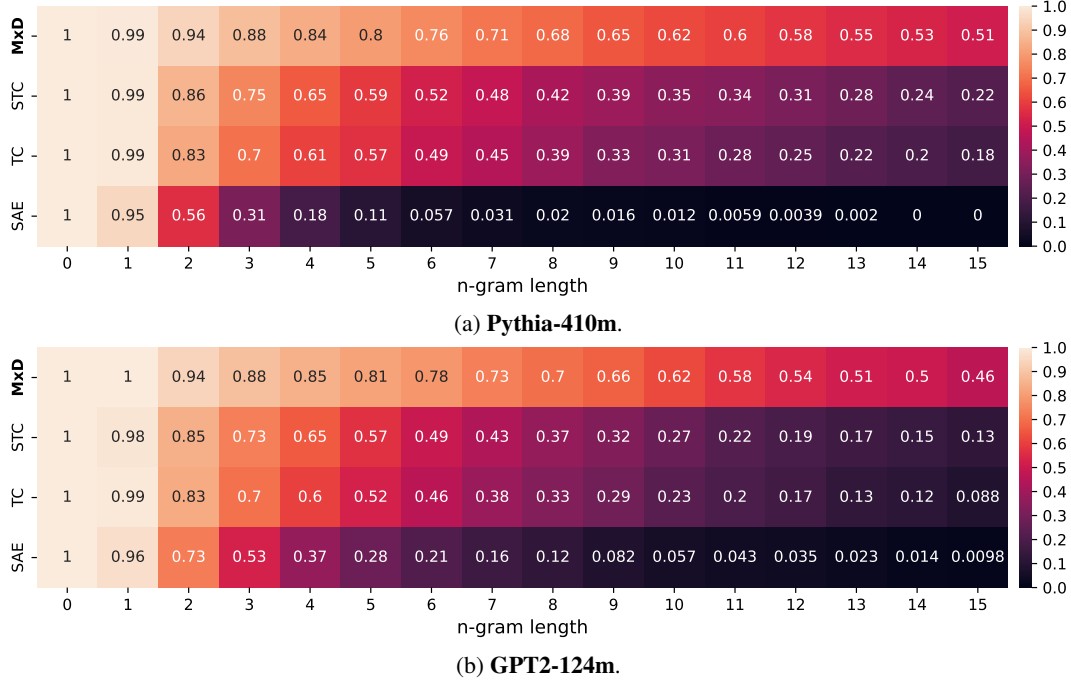

(a) **Pythia-410m**.

(b) **GPT2-124m**.

Figure 4: Proportion of 512 generated samples that contain $n$ predicted future words identical to the original model's output when replacing the base LLM's MLP layer with the sparse layers.

## 3.2 Feature evaluations

The accurate reconstruction of MxD models in Section 3.1 provides some evidence that MxDs are faithfully emulating the original MLP layers' functional mapping. However, for interpretability, we care equally about the extent to which the learned features correspond to specialized, human-interpretable concepts. We confirm that MxD's features compete with the baselines quantitatively in

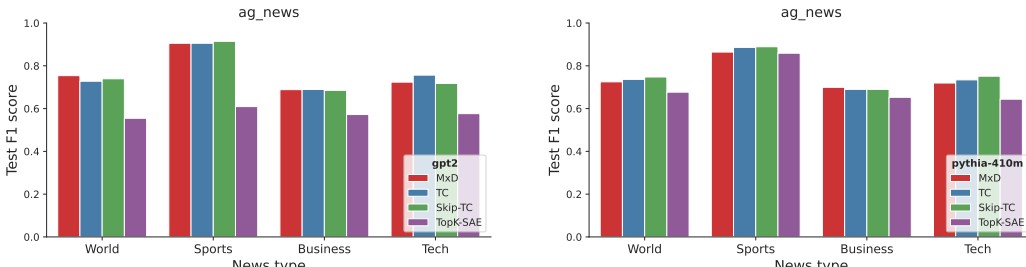

Figure 5: Highest F1 score probing for 'news category' [48] on individual features/experts. As expected, the MxDs remain competitive with the Transcoder baselines, outperforming TopK-SAEs.

two ways: through probing for known concepts in Section 3.2.1 and by steering the model using the learned features Section 3.2.2. For all experiments in this section, we use the $K = 32$ models.

**Shared experts and specialization**  Interestingly, we find MxDs naturally learn a 'shared' expert performing a common base transformation–the remaining $K - 1$ active experts are thus free to dedicate their capacity to modelling features unique to individual tokens. This emergent shared/private processing complements recent trends to use shared experts *by design* in MoEs [43, 44, 45, 46, 47] with [43] arguing this facilitates greater specialization. Furthermore, one may view the skip connection in STCs [26] as performing an analogous role to the shared expert. With MxDs, however, *all* units have the same high capacity to accurately learn separate subcomputation regardless of the frequency or rarity of features.

We also observe that our trained MxDs exhibit very few 'dead' experts, as shown in Appendix C.1, with many experts contributing actively. Furthermore, initial ablations in Appendix C.2 show that one can train MxDs without shared experts if desired, at small performance cost. Please see qualitative results of activated tokens for particular experts in Appendix E.

### 3.2.1 Sparse probing with individual features/experts

One challenge is that the sparse layers learn features in an unsupervised manner. As pointed out in [23], we therefore do not know which high-level features we ought to expect the model to learn (or even whether they exist in the OpenWebText training data). Nonetheless, we can reasonably expect a useful unsupervised model to learn at least a handful of commonly occurring concepts and linguistic themes. We accordingly focus our evaluation on the relative abilities of the sparse models to learn features well-predicting a variety of binary features used in the literature.

Concretely, to quantify the extent to which sparse layer features reliably fire in response to common high-level, interpretable concepts of natural language, we adopt the experimental settings of [49, 23, 19], training binary probes on the individual units of specialization (sparse hidden units $z_n$ for TCs/SAEs and expert units $a_n$ for MxDs–all pre-activation). For probing of sample-level concepts, we mean-pool activations across all non-padding tokens [19]. We train separate probes on $100$ features with the largest mean difference between positive and negative activations, as per [49].

We perform experiments on all $24$ binary probing tasks in the SAEBench suite [19]. Four of which are shown in Figure 5, plotting the best F1 score (on a held-out set) for news topic classification in a 1-vs-all setting [48]. As can be seen, there exist individual MxD expert units that are predictive of various categories of news articles, competitive with the baselines. We refer readers to Appendix B.7 for additional experiments on 20 more sample-level probing tasks, 10 token-level probing tasks, and experimental details.

### 3.2.2 Feature steering

Specific features might reliably fire in response to interpretable patterns of the input, yet not contribute to the generation process. Here, we aim to test this functional role of features by steering the LLMs. We note that these experiments do not aim to establish TCs/MxDs as competitive with the SOTA for controllable LLM generation. Rather, we aim to validate that the learned features contribute mechanistically to the LLM's forward pass in a predictable way.

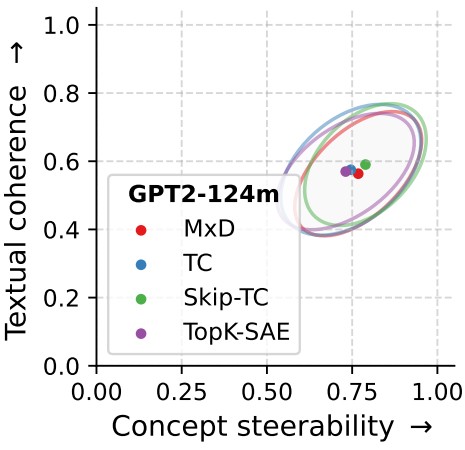 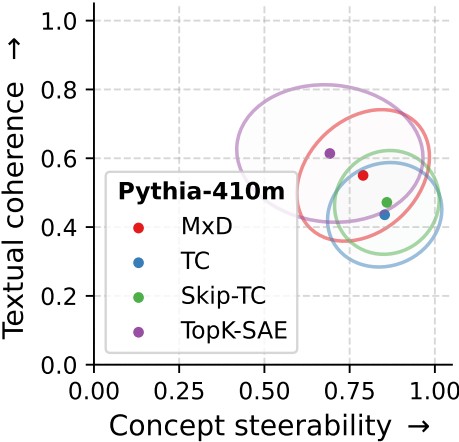

Figure 6: Mean score along dimensions of 'textual coherence' and 'steerability' of text generated by steering with the first 100 features of the sparse layers. Each sample is scored by 2 LLM judges.

**Mechanisms for steering**  Let $\lambda \in \mathbb{R}$ be a hyperparameter controlling the desired 'strength' of the model edit. For TCs, we hook the forward pass at the relevant layer to increase the presence of target feature $n$ with $\hat{\mathbf{y}} = \mathbf{y} + \lambda \mathbf{d}_n$. In contrast, MxDs can be steered with $\hat{\mathbf{y}} = \mathbf{y} + \lambda \cdot (\mathbf{W}_n^\top \mathbf{z})$. Intuitively, increasing the weight of an expert's contribution in the forward pass modulates the token representation in the direction of the learned specialization.

**Results**  We perform steering with the first 100 neurons/experts individually, using $\lambda := 100$ for all experiments. We generate a collection of 10 synthetic outputs for each neuron, each string consisting of 32 generated tokens to the prompt ''Let's talk about ''. We then ask two LLMs[2] to rate the collection of text along two dimensions separately: (1) the extent to which a shared concept, theme, or linguistic pattern is present throughout the generated collection of text, and (2) the grammatical fluency of the text (please see Appendix D.1 for the full prompt). As can be seen from the mean scores over the 100 neurons shown in Figure 6, MxDs are competitive with the baselines, exhibiting a similar trade-off between textual coherence and presence of concept as we expect.

## 4   Related work

**Sparse decompositions**  Learning sparse [50, 25], non-negative [51] features of a data signal has found many applications in computer vision [15, 52, 53, 54] and natural language processing [55, 56, 57], motivated by the pursuit of interpretable, parts-based representations [13, 14]. In transformer-based language models [3], similar variants have been proposed for post-hoc analysis; sparse autoencoders (SAEs) are a popular method that rewrites latent features as non-negative combinations of atoms in a learned overcomplete dictionary, imposing either soft sparsity penalties [6, 22, 31] or thresholding activations directly [23, 58, 59]. Recent work aims to sparsify the existing layers of pretrained LLMs, either by learning new MLPs with sparse hidden units [29] (for circuit analysis [27] or more interpretable yet faithful computation [26, 60]), or by decomposing model parameters directly using attribution [61] and/or masking [62]. Despite the surge of interest in SAEs, many works are emerging drawing attention to their limitations–underperforming baselines for probing [63], unlearning [64], and steering [65], in addition to other pathologies [66, 32, 67, 68].

**Conditional computation**  One natural alternative to static fully connected layers is conditional computation [69, 70]. Tracing back to the early work of [71, 72], single dense layers are replaced with specialized subunits–conditional on the input–as a form of layer-level sparsity. The Mixture of Experts (MoE) architecture [36, 73, 74] is a prominent example of conditional computation, breaking the link between parameter count and FLOPs. Consequently, MoEs have seen rapid adoption in SOTA models in recent years–scaling to very large parameter counts [75, 76, 77, 78, 79]. For

---

[2]We use `gemini-2.0-flash` and `llama-4-scout-17b-16e-instruct` as two independent LLM judges.

parameter-efficient instruction tuning [37] introduces conditional (IA)$^3$ adapters [38], modulating the MLP hidden dimension with the Hadamard product. Our proposed formulation with factorized weight tensors yields 'MoVs' [37] as a less scalable special case (see Appendix A.5). In contrast, MxDs model the decoder output space directly for reconstruction, and also provide significantly more specialized units than [37], making MxDs more suitable for our goal of interpretability.

Whilst the primary focus of MoEs has been on their impressive capabilities, the literature has observed that individual experts often specialize in particular semantic patterns of the input data, despite not being trained to do so [80, 81, 43, 82, 83]. For example, many works find that data that are in some sense similar are routed to the same experts–specializing to object shapes [84], texture [85], image category [86], or semantic patterns in natural language [36]. In the context of large language models, this emergent property of specialization in MoEs has been a primary focus of recent work: from encouraging monosemantic experts [87] or sparsity amongst experts' weights [88] to efficiently scaling the expert count for fine-grained specialization [40]. In contrast to these works exploring pre-training, we explore an efficient design of MoE to replace existing LLMs' dense layers.

**Interpretability by design**  Whilst MxDs follow the paradigm of specialization-via-sparsity, a number of promising interpretable mechanisms have been recently proposed that do not rely on sparsity constraints. For example, recent work explores bilinear layers [89], combinatorial structure [90], and/or tensor networks [91] as alternative ways of designing interpretable architectures. Notably, [89] show how the Hadamard product between two linear transformations of the same input vector allows direct interpretation (such as analyzing input-output feature interactions). Whilst MxDs yield a similar Hadamard product forward pass, it involves two different input vectors, and establishes a theoretical equivalence to mixture of experts and conditional computation when sparsity is present in one of the operands (i.e., the expert coefficients).

# 5   Conclusion

In this paper, we showed the benefits of decomposing dense layers' computations as a mixture of interpretable sublayers. We proposed the Mixture of Decoders (MxD) layer to achieve this at scale, proving that MxDs' linear experts preserve the matrix rank properties of the original decoders. Experimentally, we showed MxDs significantly outperform on the sparsity-accuracy frontier when trained to replace dense MLP layers. Quantitative results on sparse probing and feature steering demonstrated MxDs nonetheless learn specialized latent features similarly to existing interpretability techniques. Crucially, MxDs reexamine the dominating neuron-level sparsity paradigm of popular techniques, providing evidence that specialization doesn't have to come with such a high cost to model performance. We believe MxDs (and specialization at the layer-level more generally) are an important step towards sparsity without sacrifice, and hope future work continues to build interpretable mechanisms that better preserve model capabilities. We are excited about future work exploring MxDs (and mixtures of linear transformations more generally) in alternative settings, such as for cross-layer features or transformations [92].

**Limitations**  Our experiments show MxDs outperform on the sparsity-accuracy frontier on 4 diverse LLMs. Whilst we fully anticipate this trend to scale just as well as with sparse MLPs in even larger models, our experiments only provide direct evidence for LLMs with up to 3B parameters, given our limited resources.  Furthermore, whilst the TopK activation can greatly reduce the decoders' FLOPs, the large encoders in sparse MLPs and the gating function in MxDs remain an additional inference-time cost. Future work could explore hierarchical structures [87, 36] and/or efficient retrieval [93] for further reductions in FLOPs. Secondly, MoEs are prone to issues of expert imbalance [73], or collapse [94].  Through random weight initialization alone, we find MxDs in our experiments learn diverse experts without collapsing to the base decoder. However, we expect standard MoE load-balancing [95] or diversity losses [96] to be useful for MxDs should one need more explicit ways of encouraging expert diversity, or when training new interpretable MxD architectures end-to-end. With regards to feature evaluations, our steering experiments rely on LLMs as judges. Whilst this is commonplace in recent popular steering benchmarks [65], there is mixed evidence emerging about the reliability of all base models [97, 98]. Our experiments attempt to circumvent issues by reporting scores across two capable SOTA models, but we caution inferring too much into the absolute values of reported scores (rather than the relative performance between sparse layers).

# Acknowledgments

**James Oldfield** is grateful to Demian Till for reviewing the draft and providing valuable feedback and suggestions, and would also like to thank Markos Georgopoulos, Benjamin Hayum, and Wisconsin AI Safety Initiative's Safety Scholars for insightful discussions throughout the project. We are also grateful to the open-source Zulip platform for facilitating research discussion. **Sharon Li** is supported in part by the AFOSR Young Investigator Program under award number FA9550-23-1-0184, National Science Foundation under awards IIS-2237037 and IIS-2331669, Office of Naval Research under grant number N00014-23-1-2643, Schmidt Sciences Foundation, Open Philanthropy, Alfred P. Sloan Fellowship, and gifts from Google and Amazon. **Shawn Im** is also supported by the National Science Foundation Graduate Research Fellowship Program under Grant No. 2137424. Any opinions, findings, and conclusions or recommendations expressed in this material are those of the author(s) and do not necessarily reflect the views of the National Science Foundation. Support was also provided by the Graduate School and the Office of the Vice Chancellor for Research at the University of Wisconsin-Madison with funding from the Wisconsin Alumni Research Foundation. **Mihalis Nicolaou** is supported in part by the TensorICE project (EXCELLENCE/0524/0407), implemented under the social cohesion programme "THALIA 2021-2027", co-funded by the European Union through the Research and Innovation Foundation.

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

# Appendix

## Table of Contents

## A   Proofs and additional technical results

### A.1   Proof of rank equality

*Proof of Lemma 1.* We first derive the expression for expert $n$'s weight matrix $\mathbf{W}_n = \mathbf{D}\operatorname{diag}(\mathbf{c}_n) \in \mathbb{R}^{H \times O}$ and then show the rank equality that follows. First, recall that we have the third-order weight tensor defined as

$$\boldsymbol{\mathcal{W}}(n, h, :) = \mathbf{c}_n * \mathbf{d}_h \in \mathbb{R}^O,$$

for matrices $\mathbf{C} \in \mathbb{R}^{N \times O}$, $\mathbf{D} \in \mathbb{R}^{H \times O}$. We can express each element of the tensor $\boldsymbol{\mathcal{W}} \in \mathbb{R}^{N \times H \times O}$ in terms of elements of the two matrices as

$$\boldsymbol{\mathcal{W}}(n, h, o) = c_{no} \cdot d_{ho} = (\mathbf{D})_{ho} \cdot c_{no}. \tag{6}$$

Equation (6) shows that for a fixed expert $n$, the $n^{\text{th}}$ row $\mathbf{c}_n \in \mathbb{R}^O$ essentially scales the columns of matrix $\mathbf{D} \in \mathbb{R}^{H \times O}$. This is equivalent to right-multiplying matrix $\mathbf{D}$ by a diagonal matrix formed from $\mathbf{c}_n \in \mathbb{R}^O$. Indeed, the $(h, o)$ entry of such matrix product is

$$[\mathbf{D}\operatorname{diag}(\mathbf{c}_n)]_{ho} = \sum_{i=1}^{O} (\mathbf{D})_{hi}\operatorname{diag}(\mathbf{c}_n)_{io} \tag{7}$$

$$= (\mathbf{D})_{ho}\operatorname{diag}(\mathbf{c}_n)_{oo} \tag{8}$$

$$= d_{ho} \cdot c_{no}, \tag{9}$$

since all off-diagonal terms (i.e., $i \neq o$) in Equation (7) vanish and $\operatorname{diag}(\mathbf{c}_n)_{oo} = c_{no}$ by construction. Comparing Equation (6) and Equation (9) shows that, for every $h \in \{1, 2, \ldots, H\}$ and $o \in \{1, 2, \ldots, O\}$ we have

$$\boldsymbol{\mathcal{W}}(n, h, o) = [\mathbf{D}\operatorname{diag}(\mathbf{c}_n)]_{ho}.$$

Hence, indexing into the first mode of the tensor alone gives us the matrix-valued expression for expert $n$ as claimed:

$$\boldsymbol{\mathcal{W}}(n, :, :) = \mathbf{W}_n = \mathbf{D}\operatorname{diag}(\mathbf{c}_n) \in \mathbb{R}^{H \times O}.$$

Finally, a standard result in linear algebra [99] has that $\operatorname{rank}(\mathbf{AB}) = \operatorname{rank}(\mathbf{A})$ for any $\mathbf{A} \in \mathbb{R}^{H \times O}$ and invertible matrix $\mathbf{B} \in \mathbb{R}^{O \times O}$. Since matrix $\operatorname{diag}(\mathbf{c}_n) \in \mathbb{R}^{O \times O}$ is invertible by assumption in Lemma 1, setting $\mathbf{A} = \mathbf{D}$ and $\mathbf{B} = \operatorname{diag}(\mathbf{c}_n)$ yields the rank equality. $\qquad\square$

## A.2 Proof of MxD forward pass equivalence

Recall we have input vector $\mathbf{z} \in \mathbb{R}^H$, expert coefficients $\mathbf{a} \in \mathbb{R}^N$, and layer weights $\boldsymbol{\mathcal{W}} \in \mathbb{R}^{N \times H \times O}$. The weights are defined in Equation (4) element-wise through the Hadamard product $*$ as

$$\boldsymbol{\mathcal{W}}(n, h, :) = \mathbf{c}_n * \mathbf{d}_h \in \mathbb{R}^O, \quad \forall n \in \{1, \ldots, N\}, \ h \in \{1, \ldots, H\},$$

for learnable parameters $\mathbf{C} \in \mathbb{R}^{N \times O}$, $\mathbf{D} \in \mathbb{R}^{H \times O}$. Lemma 2 states that MxD's forward pass can be equivalently expressed as

$$\sum_{n=1}^{N} a_n(\mathbf{W}_n^\top \mathbf{z}) = (\mathbf{C}^\top \mathbf{a}) * (\mathbf{D}^\top \mathbf{z}).$$

*Proof of Lemma 2.* The LHS can first be re-written as an explicit sum over the hidden dimension

$$\hat{\mathbf{y}} = \sum_{n=1}^{N} a_n(\mathbf{W}_n^\top \mathbf{z}) = \sum_{n=1}^{N} \sum_{h=1}^{H} a_n(\mathbf{w}_{nh:} z_h) \in \mathbb{R}^O. \tag{10}$$

Plugging in the definition of $\mathbf{w}_{nh:} \in \mathbb{R}^O$ from Equation (4) then yields

$$\hat{\mathbf{y}} = \sum_{n=1}^{N} \sum_{h=1}^{H} a_n(\mathbf{w}_{nh:} z_h) \tag{11}$$

$$= \sum_{n=1}^{N} \sum_{h=1}^{H} a_n\big((\mathbf{c}_n * \mathbf{d}_h) z_h\big) \tag{12}$$

$$= \left(\sum_{n=1}^{N} a_n \mathbf{c}_n\right) * \left(\sum_{h=1}^{H} z_h \mathbf{d}_h\right) \tag{13}$$

$$= (\mathbf{C}^\top \mathbf{a}) * (\mathbf{D}^\top \mathbf{z}), \tag{14}$$

which is exactly the RHS of Equation (5), showing the MxD forward pass is equivalent to the Hadamard product of $\mathbf{C}^\top \mathbf{a}$ and $\mathbf{D}^\top \mathbf{z}$. $\qquad\square$

## A.3 Intuition for weight parameterization through the lens of tensor methods

A second complementary way of viewing the MxD layer's parameterization (and its full-rank properties) is through the lens of tensor methods [34]. A tensor-based motivation for MxD's weight tensor parameterization and forward pass is presented in Appendix A.3.1 and Appendix A.3.2, respectively.

**Notation and definitions** A brief primer is first included below, based on [34] (and can be safely skipped for those already familiar):

- The **mode-$n$ fibers** of an $N^{\text{th}}$ order tensor $\boldsymbol{\mathcal{X}} \in \mathbb{R}^{I_1 \times I_2 \times \cdots \times I_N}$ are the $I_n$-dimensional column vectors obtained by fixing every index except that of the $n^{\text{th}}$ mode (e.g., $\mathbf{x}_{:i_2 i_3} \in \mathbb{R}^{I_1}$ are the mode-1 fibers of a third-order tensor $\boldsymbol{\mathcal{X}} \in \mathbb{R}^{I_1 \times I_2 \times I_3}$). Stacking all mode-$n$ fibers column-wise yields the so-called **mode-$n$ unfolding** $\mathbf{X}_{(n)} \in \mathbb{R}^{I_n \times \tilde{I}_n}$, with number of columns given by the product of remaining dimensions $\bar{I}_n = \prod_{\substack{t=1 \\ t \neq n}}^{N} I_t$.

- The **Khatri-Rao product** (denoted by $\odot$) between two matrices $\mathbf{A} \in \mathbb{R}^{I \times K}$ and $\mathbf{B} \in \mathbb{R}^{J \times K}$, is the column-wise Kronecker product (denoted by $\otimes$):
  $\mathbf{A} \odot \mathbf{B} := \begin{bmatrix} \mathbf{a}_{:1} \otimes \mathbf{b}_{:1} & \cdots & \mathbf{a}_{:K} \otimes \mathbf{b}_{:K} \end{bmatrix} \in \mathbb{R}^{(I \cdot J) \times K}$.

- The **mode-$n$ (vector) product** of a tensor $\boldsymbol{\mathcal{X}} \in \mathbb{R}^{I_1 \times I_2 \times \cdots \times I_N}$ with a vector $\mathbf{u} \in \mathbb{R}^{I_n}$ is denoted $\boldsymbol{\mathcal{X}} \times_n \mathbf{u}$ and has entries $(\boldsymbol{\mathcal{X}} \times_n \mathbf{u})_{i_1 \ldots i_{n-1} i_{n+1} \ldots i_N} = \sum_{i_n=1}^{I_n} x_{i_1 i_2 \ldots i_N} u_{i_n}$.

### A.3.1  MxD weight tensors through the Khatri-Rao product

MxDs construct the collective weight tensor through the Khatri-Rao product $\odot$ [34] of the two factor matrices $\mathbf{C} \in \mathbb{R}^{N \times O}$, $\mathbf{D} \in \mathbb{R}^{H \times O}$. Concretely, the mode-3 unfolding[3] of the third-order weight tensor $\boldsymbol{\mathcal{W}} \in \mathbb{R}^{N \times H \times O}$ in MxDs from Equation (4) is alternatively given by:

$$\mathbf{W}_{(3)} := (\mathbf{C} \odot \mathbf{D})^\top \in \mathbb{R}^{O \times (N \cdot H)}. \tag{15}$$

Given that the factor matrices are learned end-to-end without constraints, they are likely of full column-rank, i.e. $\text{rank}(\mathbf{D}) = \text{rank}(\mathbf{C}) = O$ (as $N > O$, $H = 4 \cdot O > O$ in practice given the MLP layers' larger bottleneck). Consequently, their Khatri-Rao product parameterizing the collective $N$ experts' weights will be of maximum rank $O$ too, through Lemma 1 of [100]. As a result, parameterized this way, the $O$-dimensional fibers likely span the full output space.

### A.3.2  Tensorized MxD forward pass

Furthermore, the layer's forward pass can then be viewed as performing two tensor contractions between the third-order weight tensor $\boldsymbol{\mathcal{W}} \in \mathbb{R}^{N \times H \times O}$ (collecting all $N$ experts' $H \times O$-dimensional matrices) and expert coefficients $\mathbf{a} \in \mathbb{R}^N$ and hidden activations $\mathbf{z} \in \mathbb{R}^H$. This can be expressed in terms of the so-called mode-$n$ product (denoted by $\times_n$) [34] as follows:

$$\begin{aligned}
\hat{\mathbf{y}} &= \sum_{n=1}^{N} a_n \cdot \left( \mathbf{W}_n^\top \mathbf{z} \right) \\
&= \sum_{n=1}^{N} a_n \sum_{h=1}^{H} \mathbf{w}_{nh} z_h = \sum_{n=1}^{N} \sum_{h=1}^{H} a_n z_h \mathbf{w}_{nh} \\
&= \boldsymbol{\mathcal{W}} \times_1 \mathbf{a} \times_2 \mathbf{z} \in \mathbb{R}^O.
\end{aligned} \tag{16}$$

### A.4  GLU encoders are a mixture of rank-1 linear experts

Both the proposed MxDs and Gated Linear Units (GLUs) [33] share a similar functional form, using the element-wise product. However, there are crucially important differences between GLUs and MxDs that make both their interpretation and model capacity different.

In short, the technical results here in our paper show that GLUs' encoder can be viewed as a linear mixture of expert layer with rank-1 experts. Furthermore, GLUs can be modified and extended to MxDs with two additions to their model form as detailed at the end of this subsection. First, recall that the GLU encoder [33] computes:

$$\mathbf{y}_{\text{GLU}} = \psi(\mathbf{E}_{\text{GLU}}^\top \mathbf{x}) * \left( \mathbf{E}^\top \mathbf{x} \right) \in \mathbb{R}^H, \tag{17}$$

for input vector $\mathbf{x} \in \mathbb{R}^I$, learnable weights $\mathbf{E}_{\text{GLU}}, \mathbf{E} \in \mathbb{R}^{I \times H}$, and activation function $\psi(.)$. To transform Equation (17) into the same model form as MxDs, we first pre-multiply the LHS by the identity matrix to match the MxD model form of Equation (5), yielding:

$$\mathbf{y}_{\text{GLU}} = \left( \mathbb{I}^\top \mathbf{a} \right) * \left( \mathbf{E}^\top \mathbf{x} \right), \tag{18}$$

---

[3]which is simply a reshaping of a higher-order tensor into a matrix, arranging all $N$ expert matrices' column vectors along the columns of a new matrix.

where $\mathbf{a} = \psi(\mathbf{E}_{\text{GLU}}^{\top}\mathbf{x}) \in \mathbb{R}^H$ and $\mathbb{I} \in \mathbb{R}^{H \times H}$ is the $H$-dimensional identity matrix. Next, we can write this explicitly in terms of a linear MoE with expert weights $\mathbf{W}_n \in \mathbb{R}^{I \times H}$ as follows:

$$\mathbf{y}_{\text{GLU}} = \left(\mathbb{I}^{\top}\mathbf{a}\right) * \left(\mathbf{E}^{\top}\mathbf{x}\right) \tag{19}$$

$$= \sum_{n=1}^{H} a_n\left(\mathbf{W}_n^{\top}\mathbf{x}\right) \tag{20}$$

$$= \sum_{n=1}^{H} a_n\left(\mathbf{E}\operatorname{diag}\left((\mathbb{I})_n\right)\right)^{\top}\mathbf{x}\right), \tag{21}$$

where $(\mathbb{I})_n \in \mathbb{R}^H$ is the $n^{\text{th}}$ row of the $H$-dimensional identity matrix (i.e. a one-hot vector with its only non-zero element at index $n$). We draw particular attention to how the $n^{\text{th}}$ expert's matrix $\mathbf{W}_n = \mathbf{E}\operatorname{diag}\left((\mathbb{I})_n\right) \in \mathbb{R}^{I \times H}$ essentially picks out the $n^{\text{th}}$ column of $\mathbf{E}$, leaving all remaining $H-1$ columns as zero vectors. **Therefore, GLU encoders compute a MoE with linear expert weights of (at most) rank 1**. This relationship between GLUs and conditional computation is consistent with prior work interpreting individual GLU column vectors as experts [101]. Whilst GLUs' encoders' model form does not put any inherent restrictions on the total number of rank-1 terms that can contribute to the output, the sparsity necessary for specialization does.

We conclude this section by summarizing the two technical changes needed to transform GLUs into full-rank linear MoEs based on the Hadamard product:

1. Replace $\mathbb{I}$ in Equation (18) with learnable, non-diagonal weight matrices for full-rankness.

2. Choose $\psi(.)$ to produce non-negative, sparse coefficients to encourage specialization through sparsity among the experts (for example, a `softmax` function, or a `ReLU` activation followed by `TopK`).

The first of the steps above provides full-rankness, whilst the second brings the sparsity and non-negativity needed for specialization. We include a notebook showing this connection in PyTorch at: https://github.com/james-oldfield/MxD/blob/main/glus-to-moes.ipynb.

## A.5 Hadamard-factorized tensors generalize MoVs

Prior work [37] proposes to linearly combine $N$ many $(\text{IA})^3$ adapters [38] for parameter-efficient MoEs for instruction fine-tuning. The implementation results in a very similar functional form to the factorized forward-pass in MxDs. Interestingly, the Hadamard product parameterization of the third-order weight tensor in Equation (4) provides a more general framework through which one can also derive MoVs' model form, shedding light on the relationship to the proposed MxDs and their benefits. Concretely, factorizing the weight tensor instead along the *second* mode as $\mathcal{W}(n,:,o) = \mathbf{c}_n * \mathbf{d}_o \in \mathbb{R}^H$ in our framework immediately recovers MoV [37] as a special case. In particular, in contrast to the MxD in Appendix A.3 whose weight tensor can be parametrized equivalently through its mode-3 unfolding [34], MoV's implicit weight tensor can be given in terms of its mode-2 unfolding in terms of a similar Khatri-Rao product of two factor matrices.

Instead, MoVs in analogy would yield expert weights by pre-multiplying $\mathbf{D}$ as: $\mathbf{W}_n = \operatorname{diag}(\mathbf{c}_n)\,\mathbf{D} \in \mathbb{R}^{H \times O}$ for much larger $\mathbf{C} \in \mathbb{R}^{N \times H}$). Due to $H \gg O$, **our proposed MxD formulation yields around $4\times$ the number of specialized units as MoVs** with the same parameter budget (yet MoVs' experts are of no higher rank than MxDs'), making MxDs a much more suitable and efficient class of layer for our goal of scalable specialization. We therefore see that the proposed lens of tensor methods for unification provides valuable insights about how to design more interpretable layers with the minimum trade-off to capabilities.

## A.6 TopK activation function

Formally, for activations $\mathbf{z} \in \mathbb{R}^H$ (possibly with the ReLU activation applied), the TopK activation function can be formulated elementwise as the following:

$$\text{TopK}(\mathbf{z})_h = \begin{cases} z_h, & \text{if } z_h \geq \tau_K(\mathbf{z}) \\ 0, & \text{otherwise} \end{cases},$$

where $\tau_K(\mathbf{z}) \in \mathbb{R}$ returns the value of the $K^{\text{th}}$ largest element of vector $\mathbf{z} \in \mathbb{R}^H$. Intuitively, the TopK activation retains only the largest $K \leq H$ elements of the $H$ neurons, setting the rest to zero. We note here that the definition above does not handle ties (although this is very unlikely in practice, given the activations are continuous).

# B  Additional quantitative results and ablations

## B.1  Faithfulness in output space

Experiments in Section 3.1.2 of the main paper report the number of future predicted words that are identical from the original model and from the LLM with the sparse layer replacements. We also show qualitative examples of the first 8 prompts and the subsequent 'diffs' (using Python 3's `difflib`) of the generated tokens in Figures 7 and 8–we see MxDs' superior ability to preserve model functionality as it propagates through to the output space of future tokens.

## Generation 1, Pythia-410m

```
GT:   There are times when you need to take a break from your daily routine and just
MxD:  There are times when you need to take a break from your daily routine and just
STC:  There are times when you need to be able to do something that is not easy.
TC:   There are times when you need to take a break from your daily routine and just
SAE:  There are times when you need to be in the air, but not in a high-speed
```

## Generation 2, Pythia-410m

```
GT:   Humanitarian chief warns capacity of US to be tested By Staff reports Published: Friday, May 30,
MxD:  Humanitarian chief warns capacity of US to be tested By Staff reports Published: Friday, May 30,
STC:  Humanitarian chief warns capacity of US to be tested By Associated Press Published: Tuesday, March 29,
TC:   Humanitarian chief warns capacity of US to be tested By Associated Press | January 24, 2013
SAE:  Humanitarian chief warns capacity of the U.S. military to be a threat The U.S. military is
```

## Generation 3, Pythia-410m

```
GT:   During the Trump administration, the Department of Homeland Security has been tasked with overseeing immigration enforcement.
MxD:  During the Trump administration, the Department of Homeland Security (DHS) has been tasked with protecting Americans
STC:  During the Trump administration, the Department of Justice has been accused of using "bribery" to influence
TC:   During the Trump administration, the Department of Homeland Security has been tasked with overseeing immigration enforcement.
SAE:  During the Trump administration, the president-elect's campaign was a "crisis" that could be solved by a
```

## Generation 4, Pythia-410m

```
GT:   Romanian newspaper ZF.ro cites a report from the Ministry of Interior and Security (MIS) that shows
MxD:  Romanian newspaper ZF.ro cites a report from the Ministry of Interior and Security (MIS) that shows
STC:  Romanian newspaper ZF.ro cites a report by the European Commission that shows that Romania is not
TC:   Romanian newspaper ZF.ro cites a report from the Ministry of Foreign Affairs and Trade (MFA) that
SAE:  Romanian newspaper ZF.ro cites a report that the country's police officers are not allowed to use
```

## Generation 5, Pythia-410m

```
GT:   Democratic Virginia Gov. Terry McAuliffe (D) on Tuesday said he will not seek re-election in 2020,
MxD:  Democratic Virginia Gov. Terry McAuliffe (D) on Tuesday said he will not seek re-election in 2020,
STC:  Democratic Virginia Gov. Terry McAuliffe (D) is running for president in the 2020 election, but he's
TC:   Democratic Virginia Gov. Terry McAuliffe (D) is facing a challenge from a group of Democratic state
SAE:  Democratic Virginia Gov. Terry McA. Gingis is a Democrat, but he's not the most popular governor
```

## Generation 6, Pythia-410m

```
GT:   A federal court has ruled that the Trump administration's travel ban on people from seven Muslim-majority
MxD:  A federal court has ruled that the Trump administration's travel ban on people from seven Muslim-majority
STC:  A federal court has ruled that the Trump administration's travel ban on people from seven Muslim-majority
TC:   A federal court has ruled that the Trump administration's travel ban on people from seven Muslim-majority
SAE:  A federal court has ordered a former U.S. ambassador to the United Nations, William H. Taylor,
```

## Generation 7, Pythia-410m

```
GT:   British Columbia takes in \$1.5 billion from the sale of its oil and gas reserves The
MxD:  British Columbia takes in \$1.5 billion from the sale of its oil and gas reserves The
STC:  British Columbia takes in \$1.5 billion in tax breaks The British Columbia government has announced that
TC:   British Columbia takes in the world The British Columbia government has announced that it will spend
SAE:  British Columbia takes in the third year of a plan to expand its coal-mining operations, but
```

## Generation 8, Pythia-410m

```
GT:   Earlier this month, I wrote about the upcoming release of the first episode of The Walking
MxD:  Earlier this month, I wrote about the upcoming release of the first episode of The Walking
STC:  Earlier this month, I wrote about the upcoming release of the "Sonic Boom" soundtrack. The album
TC:   Earlier this month, I wrote about the upcoming release of the "Sonic Mania" game. The game
SAE:  Earlier this month, I was invited to a conference in the city of Toronto. The event
```

Figure 7: **Pythia-410m**: The first few generated tokens from the base model ('**GT**') and the corresponding tokens from the model when the sparse layers replace the target MLP. Red denotes tokens that are removed, orange denotes newly inserted tokens, and green denotes matching tokens.

**Generation 1, GPT2-124m**

GT:  There are times when you need to be a little more careful with your food. You
MxD: There are times when you need to be on the lookout for a new job. But,
STC: There are times when you need to get out of bed and go to the bathroom.
TC:  There are times when I feel like I'm being judged. I've been told that my grades
SAE: There are times when you need to get a little bit more than your usual. You're

**Generation 2, GPT2-124m**

GT:  Humanitarian chief warns capacity to handle refugees is 'unprecedented' The UN refugee agency has warned that
MxD: Humanitarian chief warns capacity to handle refugees 'is at risk' The UN refugee agency has warned
STC: Humanitarian chief warns capacity to handle emergencies is at risk The UK government has warned that
TC:  Humanitarian chief warns capacity to handle emergencies is at risk The UK government has warned that
SAE: Humanitarian chief warns capacity to hold up The BBC's political correspondent, Peter Robinson, says the government

**Generation 3, GPT2-124m**

GT:  During the Trump administration, the White House has been working to make sure that its immigration
MxD: During the Trump administration, the White House has been working to make sure that its immigration
STC: During the Trump administration, the White House has been working to make sure that it is
TC:  During the Trump administration, the White House has been working to make sure that no one
SAE: During the Trump administration, the White House has been working on a plan to build a

**Generation 4, GPT2-124m**

GT:  Romanian newspaper ZF.ro cites a report by the European Commission that the country's government is considering
MxD: Romanian newspaper ZF.ro cites a report by the European Commission that the country's government is considering
STC: Romanian newspaper ZF.ro cites a report by the European Commission that the country's government is considering
TC:  Romanian newspaper ZF.ro cites a report by the European Commission that the EU is considering imposing
SAE: Romanian newspaper ZF.ro cites the "revelation" of the "death of the country's economy" as a key

**Generation 5, GPT2-124m**

GT:  Democratic Virginia Gov. Terry McAuliffe (D) said he would not seek re-election in 2018, but he
MxD: Democratic Virginia Gov. Terry McAuliffe (D) said he would not seek re-election in 2018, but he
STC: Democratic Virginia Gov. Terry McAuliffe (D) has said he will not support a bill that would
TC:  Democratic Virginia Gov. Terry McAuliffe (R) has said he will not seek re-election in 2018, but
SAE: Democratic Virginia Gov. Terry McAuliffe (R) has signed a bill that would allow the state to

**Generation 6, GPT2-124m**

GT:  A federal court has ordered the Department of Justice to pay \$1.5 million to a group
MxD: A federal court has ordered the Department of Justice to pay \$1.5 million to a group
STC: A federal court has ordered the Department of Justice to stop using a search warrant to
TC:  A federal court has ordered the Department of Justice to stop using a search warrant to
SAE: A federal court has ordered the Department of Justice to pay \$1.5 million to a former

**Generation 7, GPT2-124m**

GT:  British Columbia takes in \$1.5 billion annually from the federal government, but only about half of
MxD: British Columbia takes in \$1.5 billion in foreign aid annually, according to the government's latest report
STC: British Columbia takes in \$1.5 billion in foreign aid annually, according to the Canadian Council of
TC:  British Columbia takes in \$1.5 billion annually from the federal government, but it's not a big
SAE: British Columbia takes in \$1.5 million in revenue from the province's tourism industry, according to a

**Generation 8, GPT2-124m**

GT:  Earlier this month, I wrote about the state of our industry. We've been in a bit
MxD: Earlier this month, I wrote about the state of our industry. We've been in a state
STC: Earlier this month, I wrote about the state of the art in building a solid foundation
TC:  Earlier this month, I wrote about the state of the art for creating a custom 3D
SAE: Earlier this month, I was asked to write a piece about the "anti-Semitic" movement in Germany.

Figure 8: **GPT2-124m**: The first few generated tokens from the base model ('**GT**') and the corresponding tokens from the model when the sparse layers replace the target MLP. Red denotes tokens that are removed, orange denotes newly inserted tokens, and green denotes matching tokens.

## B.2    Additional reconstruction metrics

To highlight the scale of difference in the reconstructions between MxDs and the baselines, we also plot in Figure 9 the normalized MSE at the end of training for all models and LLMs. At the smallest values of $K$ (which we care about most for interpretability), **MxDs' normalized MSE is up to an order of magnitude smaller than Transcoders'**.

## B.3    Results on additional layers

We also fully train all models and baselines (with 4 different values of $K$) on different target layers for each model. The results are shown in Figure 10 for $48$ additional trained layers for the same setup in the original paper, using different colours to highlight that these are new results. As can be seen, the same trend holds: MxDs significantly outperform the baselines at small $K$ in all LLMs.

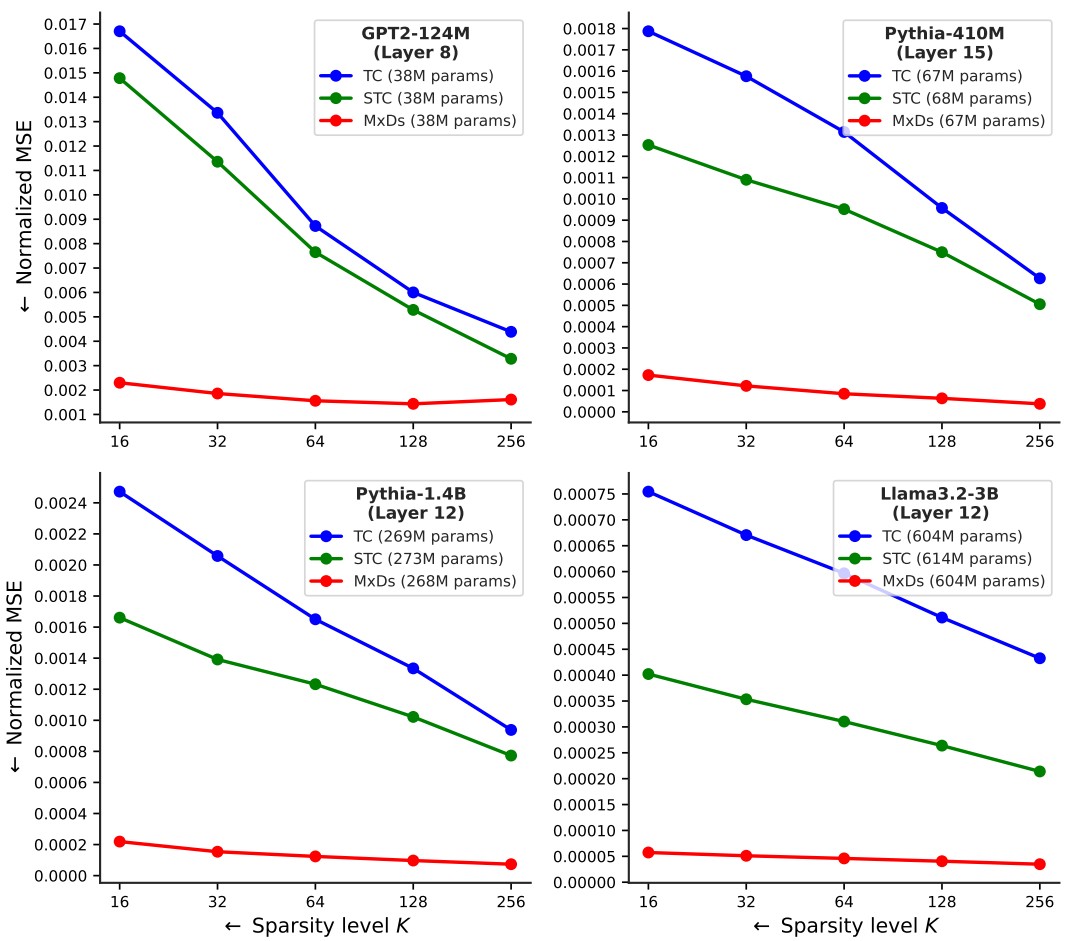

Figure 9: **Normalized MSE** at the end of training Sparse MLP layers, as a function of the number of active units (i.e., hidden neurons vs experts); with differences as large as an order of magnitude in error.

## B.4 Initial results on Gemma2-27B

In the main paper, we perform experiments on base models with up to 3B parameters. For initial evidence of scalability to even larger models, we perform here additional large-scale experiments on the 27B Gemma2 model (in half-precision, and with a smaller batch size to fit into memory). We see MxDs continue to outperform on the sparsity-accuracy frontier, as shown by the (normalized) reconstruction loss in Table 3.

Table 3: **Normalized MSE** ($\downarrow$) on Gemma2-27B, Layer 20, $K = 32$, partially trained for 50k iterations. To fit in memory, the model is loaded in half precision. We use $1/8$ the batch size and # buffers stored, and $1/2$ the multiplier on the number of latent features (with values of 4, 16, and 16, respectively).

| Iterations | 10k | 20k | 30k | 40k | 50k |
|---|---|---|---|---|---|
| Transcoder | 0.147 | 0.132 | 0.128 | 0.123 | 0.119 |
| Skip Transcoder | 0.128 | 0.107 | 0.102 | 0.100 | 0.093 |
| MxD | **0.098** | **0.096** | **0.093** | **0.086** | **0.069** |

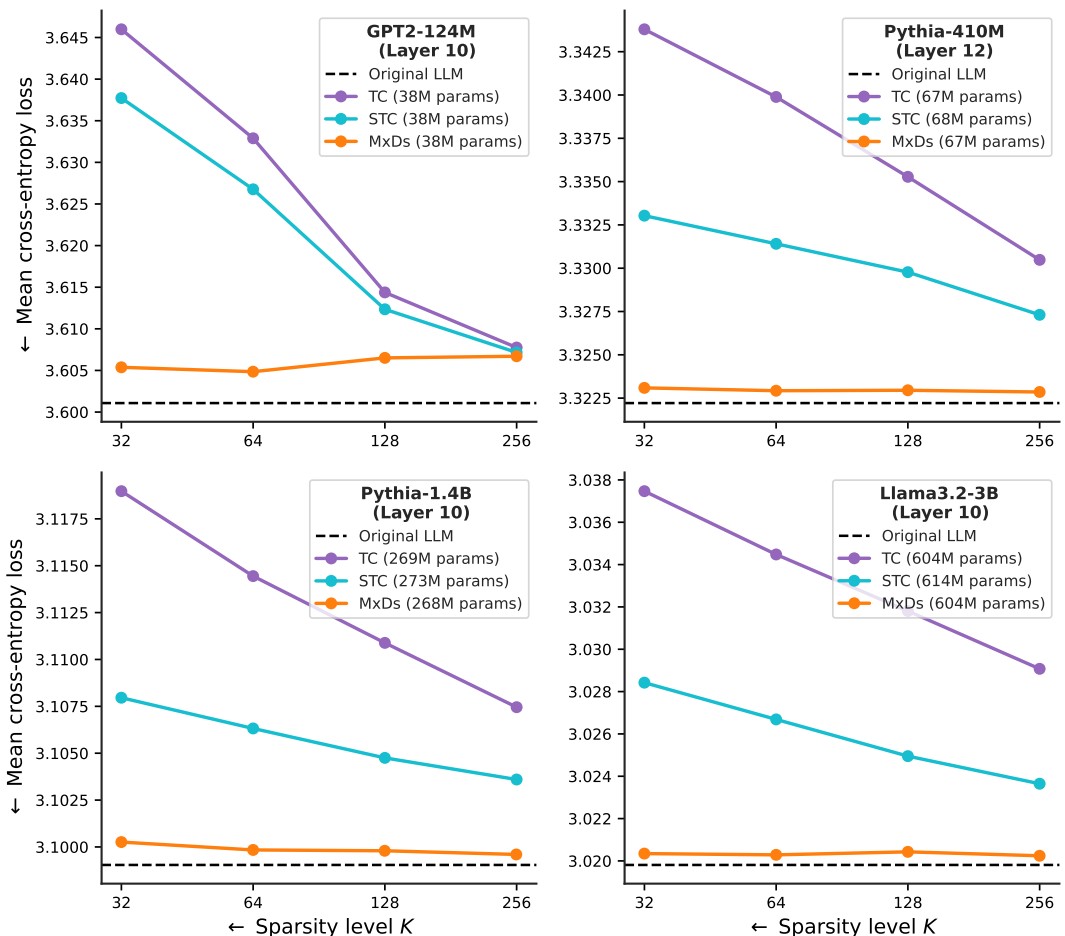

Figure 10: **Additional layer results**: model cross entropy loss preserved when replacing MLPs with Transcoders [27], Skip Transcoders [26], and MxDs, as a function of the number of active units (hidden neurons/experts). These results complement those in the main paper, but here we train a new set of additional models on different layers.

## B.5    Expert rank

This section concerns the matrix rank of the linear experts in parameter-efficient MoEs. We first compare to low-rank MoEs in Appendix B.5.1 to demonstrate the benefits of full-rankness, and then follow up in Appendix B.5.2 by confirming that the learned MxD expert ranks are close to maximum in the trained models.

### B.5.1    Comparisons to low-rank MoEs

In this section, we study the impact of expert rank on the ability of efficient MoE layers to reconstruct pre-trained MLP layers' mappings. One compelling alternative to MxDs for efficient conditional computation is the $\mu$MoE layer [40], which imposes low-rankness on expert weights to achieve parameter-efficiency. Whilst $\mu$MoEs are found to perform competitively in the pre-training setting, the impact of low-rankness on approximations of *existing* layers will determine their suitability in the sparse layer approximation setting studied in this work.

We therefore compare to $\mu$MoE layers, which we use to compute a linear MoE in place of the MLP's decoder. In CP$\mu$MoEs, $N$ experts' weight matrices are jointly parameterized through low-rank tensor structure with the CP decomposition [102, 103] for chosen rank $R \in \mathbb{N}^+$. With the same learnable encoder and expert gating matrices producing the expert coefficients $\mathbf{a} \in \mathbb{R}^N$ and hidden units $\mathbf{z} \in \mathbb{R}^H$ generated the same way as in the main paper, we train $\mu$MoE layers to approximate the

original MLP layer's output with:

$$\mu\text{MoE}(\mathbf{x}) = \sum_{n=1}^{N} \sum_{h=1}^{H} \sum_{r=1}^{R} a_n z_h \, \mathbf{D}(r,h) \cdot \mathbf{C}(r,n) \cdot \mathbf{W}(:,r) \in \mathbb{R}^O, \qquad (22)$$

where $\mathbf{C} \in \mathbb{R}^{R \times N}$, $\mathbf{D} \in \mathbb{R}^{R \times H}$, $\mathbf{W} \in \mathbb{R}^{O \times R}$ are the learnable low-rank terms of the implicit third-order tensor parameterizing all $N$ collective experts' weights.

We match the MxD experimental configuration as closely as possible for a fair comparison. For the encoders, we mirror MxDs and use the GELU activation function, which we find through ablations in Appendix B.8 to perform the best. We initialize the parameters the same as MxDs and Skip Transcoders: we use the standard PyTorch linear layer initialization for $\mathbf{D}$, $\mathbf{C}$ (and the encoder layers), and initialize $\mathbf{W}$ as the zero matrix.

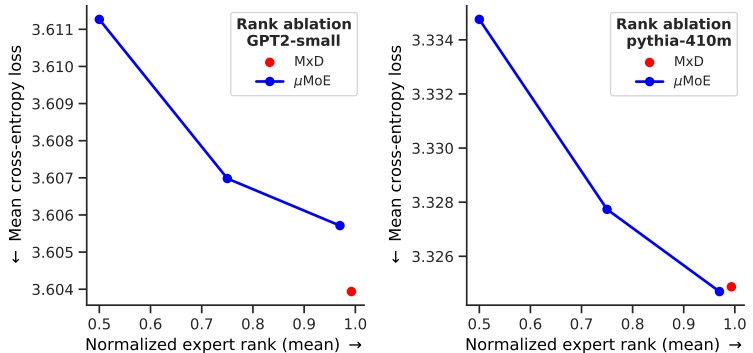

Figure 11: Comparisons to $\mu$MoEs for various choices of (normalized) rank: high rank weights best-preserve the models' downstream cross-entropy loss.

We vary the $\mu$MoE layer rank $R$, training fully 3 sparse approximation layers for $K = 32$ active experts, varying the total number of experts $N$ to keep the parameter count the same–isolating the impact of the choice of rank. As with the main experiments, we record the downstream model loss when we splice in the trained layer to replace the MLP layers, shown in Figure 11.

As can be seen, the $\mu$MoE layers perform well when they are close to full-rank (i.e. when the normalized rank $\frac{R}{O} \to 1$). Crucially, however, performance drops off notably when the rank is reduced. Whilst $\mu$MoEs still perform far better than neuron-level sparsity methods (i.e. the corresponding CE loss results in Figure 3), we observe that full-rankness is necessary for the most faithful layer approximations–which the proposed MxDs provide by design.

As a motivating example, for why SparseMoEs and SoftMoEs are not practical: SparseMoEs [36] and SoftMoEs [72] require 2.16 **trillion** parameters for a single layer, for the same 86k experts we use for `Llama-3.2-3B`. This is orders of magnitude more parameters than the entire base network itself, making it prohibitively costly for SparseMoEs to scale to sufficiently high expert counts.

### B.5.2 MxD empirical expert rank

Next, we show experimentally that the learned experts' matrices $\mathbf{W}_n = \mathbf{D}\,\text{diag}(\mathbf{c}_n) \in \mathbb{R}^{H \times O}$ are very nearly full-rank in practice, corroborating the properties of expert matrices shown theoretically in Lemma 1. We compute the mean 'normalized rank', which we take for MxDs to be the empirical matrix rank of the learned expert's weights, over the maximum possible rank given the dimensions:

$$\frac{1}{N} \sum_{n=1}^{N} \frac{\text{rank}\,(\mathbf{W}_n)}{\min\{H, O\}}. \qquad (23)$$

We show in Table 4 the normalized rank across all 4 base models: MxD's learned experts exhibit no rank deficiencies, providing further evidence of the large potential capacity of MxD layers despite their sparsity constraints on the expert-level.

Table 4: Mean normalized expert matrix rank of Equation (23) across models for the first 2k experts in $K = 32$ trained MxDs – the learned expert matrices are very close to full column rank.

| GPT2-124M | Pythia-410M | Pythia-1.4B | Llama-3.2-3B |
|---|---|---|---|
| $0.99 \pm 0.005$ | $0.99 \pm 0.007$ | $0.99 \pm 0.005$ | $0.99 \pm 0.002$ |

## B.6 Computational cost of sparse layers

We show here that there is minimal difference between the computational cost of sparse layer variants. We first report theoretical layer FLOPs, and then report empirical benchmarks.

**Parameter count & FLOPs** We first tabulate in Table 5 the theoretical parameter counts and inference-time FLOPs for MxDs vs Sparse MLPs. To be consistent with the popular PyTorch library [104] we count one fused multiply-add as one FLOP, and count the Hadamard product between two $d$-dimensional vectors as requiring $d/2$ FLOPs. Note that, For a chosen expert count $N$, we set the width of Sparse MLPs (e.g. Transcoders) to $H := N + H^*$ to parameter-match the models.

Table 5: Here $I, O$ denotes the input and output dimensions, $H^*$ the original model's hidden width, $N$ the MxD expert count, and $H$ the width of the sparse MLPs.

| | Parameter count | FLOPs |
|---|---|---|
| **Sparse MLP** | $H(I + O)$ | $H(I + O)$ |
| **MxD** | $N(I + O) + H^*(I + O)$ | $N(I + O) + H^*(I + O) + O/2$ |

**Empirical benchmarks** We next run benchmarks for a Sparse MLP layer vs MxD with a batch size of 512, and dimensions: $I = H^* = O = 1024$, and number of experts/features as $N = 8192$, $H = 9216$. The results are tabulated in Table 6, where we see performance (and cost) is similar for the two layers.

Table 6: Here $I, O$ denotes the input and output dimensions, $H^*$ the original model's hidden width, $N$ the MxD expert count, and $H$ the width of the sparse MLPs.

| | Peak memory usage (MiB) | Latency (ms) | Parameter count | Reported FLOPs ([104]) |
|---|---|---|---|---|
| **Sparse MLP** | 386.50 | 1.394 | 18,874,368 | 18,874,368 |
| **MxD** | 389.50 | 1.457 | 18,874,368 | 18,874,880 |

## B.7 Sparse probing

**Sample-level probing** Here, we follow the SAEBench [19] evaluation protocol. In this 'sample-level' setting, each text string is labeled with a binary concept at a global level (e.g., the language of the snippet, or its sentiment). This is in contrast to what we refer to as 'token-level probing', where *each token* within the text samples is labeled individually (e.g., whether a word is a certain part of speech). We perform experiments on a total of 24 sample-level sparse probing tasks with the same 'maximum mean difference' feature filtering applied in [19]. The details of the datasets used are summarized in Table 7.

**Token-level probing** We also explore sparse probing for 10 features defined at the token-level. For this, we follow [49], and include experiments training probes on the mean feature activations under tokens spanning the **surnames** of the individuals. We note that this is a significantly harder task, and makes even stronger assumptions about the features the dataset includes, but is nonetheless some additional weak evidence about the relative feature-learning abilities of the sparse models. Through various surnames, we probe for 6 occupations of individuals, whether or not individuals are alive, and individuals' labeled gender. We also experimented with probing for compound words as in [49], but found no predictive features in our trained models. Details of the surname token-level probing datasets (and the total training examples the tokenizers could parse) are included in Table 8.

Table 7: Details of sample-level sparse probing datasets used.

| Dataset | # Training examples | # Test examples | Classification task description | Number of classes |
|---|---|---|---|---|
| fancyzhx/ag_news [48] | 16,000 | 4,000 | News article topic | 4 |
| codeparrot/github-code [105] | 20,000 | 5,000 | Programming language | 5 |
| amazon_reviews_mcauley_1and5_sentiment [106] | 8,000 | 2,000 | Positive/negative review sentiment | 2 |
| Helsinki-NLP/europarl [107] | 20,000 | 5,000 | European language | 5 |
| LabHC/bias_in_bios [108] | 32,000 | 8,000 | Profession from bio | 8 |

Table 8: Details of token-level sparse probing datasets used.

| Dataset | # Training examples | # Test examples | Classification task description | Number of classes |
|---|---|---|---|---|
| Occupation [49] | 4784 | 1195 | Occupation of individual | 6 |
| Is alive? [49] | 4800 | 1199 | Are they alive | 2 |
| Gender [49] | 4800 | 1200 | Labeled gender | 2 |

**Experimental setup**  For sample-level probing, we truncate the input strings to the first 128 tokens for all datasets but for the `Github` dataset, where we take the last 128 tokens to avoid license headers [19, 49]. For token-level probing, we instead take only the last 128 tokens, where the final token contains the surname of the individual in question in the datasets of [49].

Binary probes are trained on 80% of the training data (randomly shuffled) with the `sklearn` library's `LogisticRegression` module with parameters:

- class_weight='balanced'
- penalty='l2'
- solver='newton-cholesky'
- max_iter=200

A random seed of 42 is used throughout the code to ensure reproducibility.

### B.7.1  Sparse probing results

We show in Figure 12 results on 20 additional (sample-level) sparse probing tasks, where MxDs remain competitive with the baselines. We also plot the expert activation (of the single expert with the highest F1 test set score) for the positive/negative classes for all tasks split across Figures 13 and 14. One can observe a certain degree of separability between the two semantic clusters of data given by the expert coefficient, thus confirming that individual experts are learning to specialize to particular high-level features.

We also include results on 10 token-level probing tasks in Figure 15, with the corresponding activation densities displayed in Figure 16. Whilst MxDs appear to perform slightly less well here on average, they remain competitive as expected.

### B.8  Ablations

We turn next to ablation studies to explore the value of the various model components below:

### B.8.1  Choice of sparsity constraint

We first train a variety of MxDs on GPT2 models with the TopK activation function [23] and instead train models with a ReLU followed by an explicit $\lambda ||.||_1$ sparsity penalty on the specialized components in addition to the reconstruction loss [22]. We show the results in Figure 17, where, similarly to [26], we find the TopK activation to dominate on the sparsity-accuracy frontier–we thus use the TopK activation for all experiments.

### B.8.2  Choice of MxD encoder

Secondly, we show in Figure 18 the benefits of MxDs' flexibility in inheriting the original MLP layer's encoder form/activation function. All models here are trained from scratch for the same number of tokens and with the same experimental setup as in Section 3.1, with $K = 32$. In the

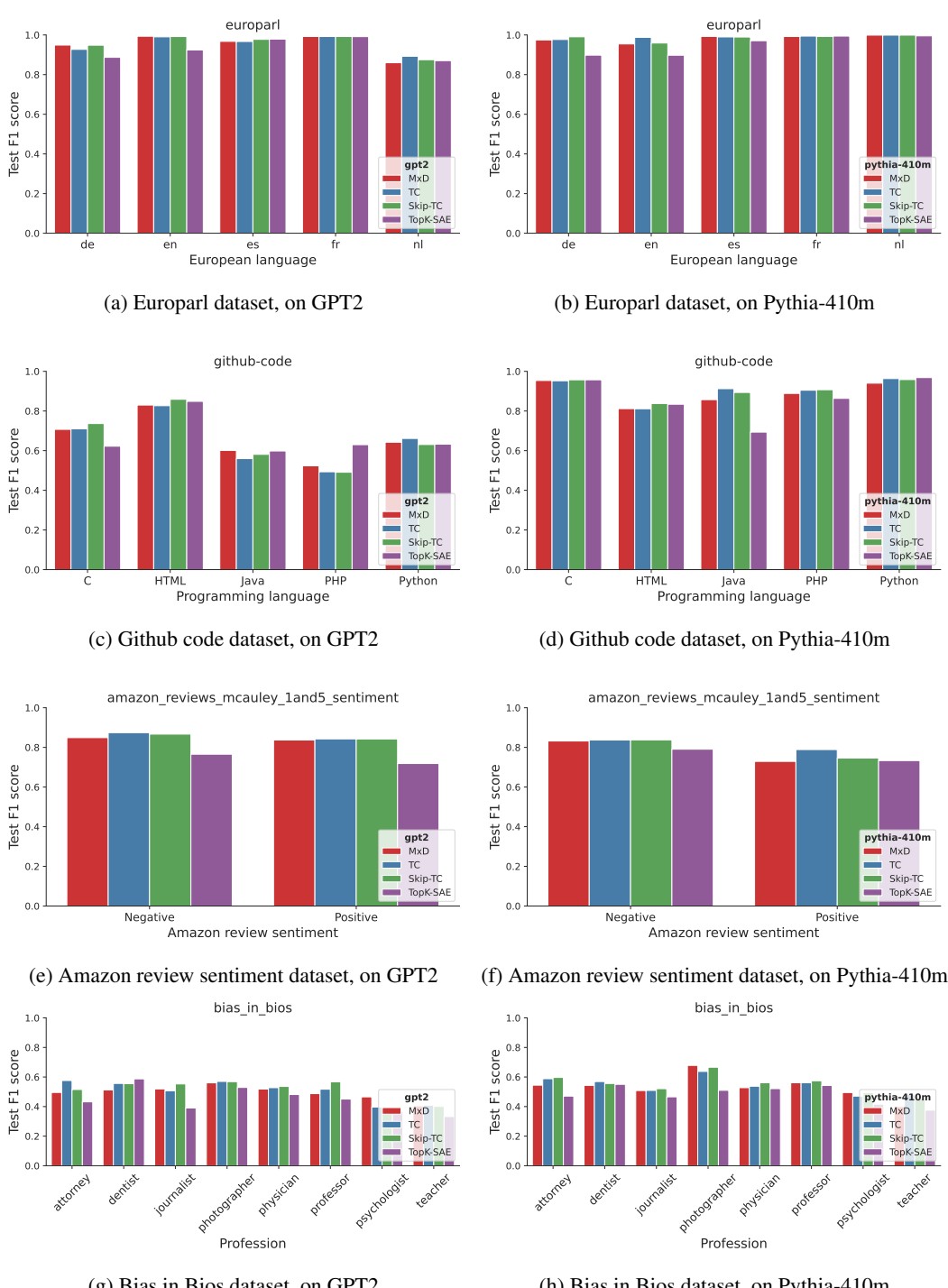

(a) Europarl dataset, on GPT2

(b) Europarl dataset, On Pythia-410m

(c) Github code dataset, on GPT2

(d) Github code dataset, on Pythia-410m

(e) Amazon review sentiment dataset, on GPT2

(f) Amazon review sentiment dataset, on Pythia-410m

(g) Bias in Bios dataset, on GPT2

(h) Bias in Bios dataset, on Pythia-410m

Figure 12: **Sample-level** sparse probing results on individual experts/features; the best F1 score on a held out set is presented.

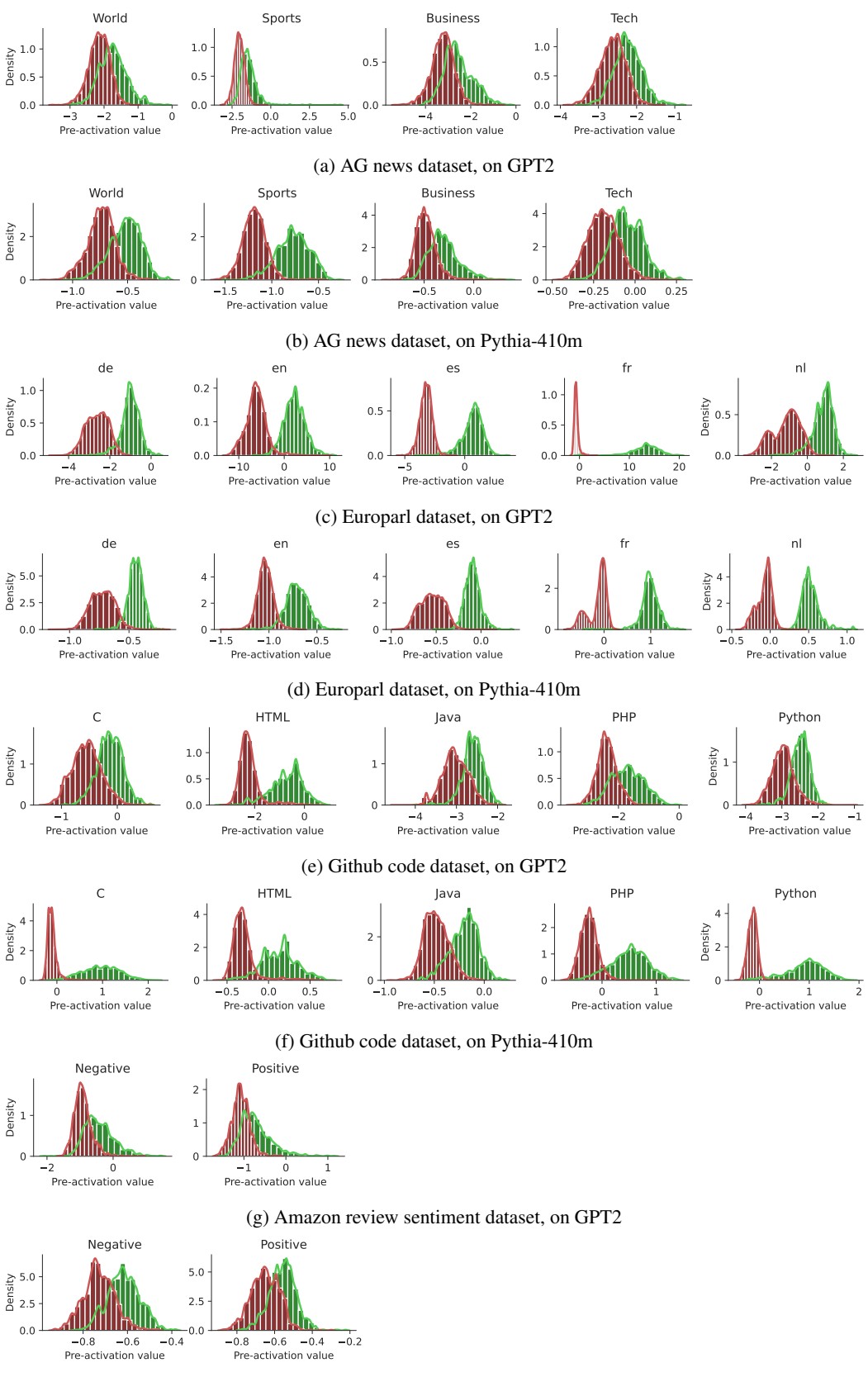

Figure 13: **[1/2] Sample-level** sparse probing results on individual experts for MxDs; here we plot the values of the expert pre-activation for **positive**/**other** classes (in the 1-vs-all setting).

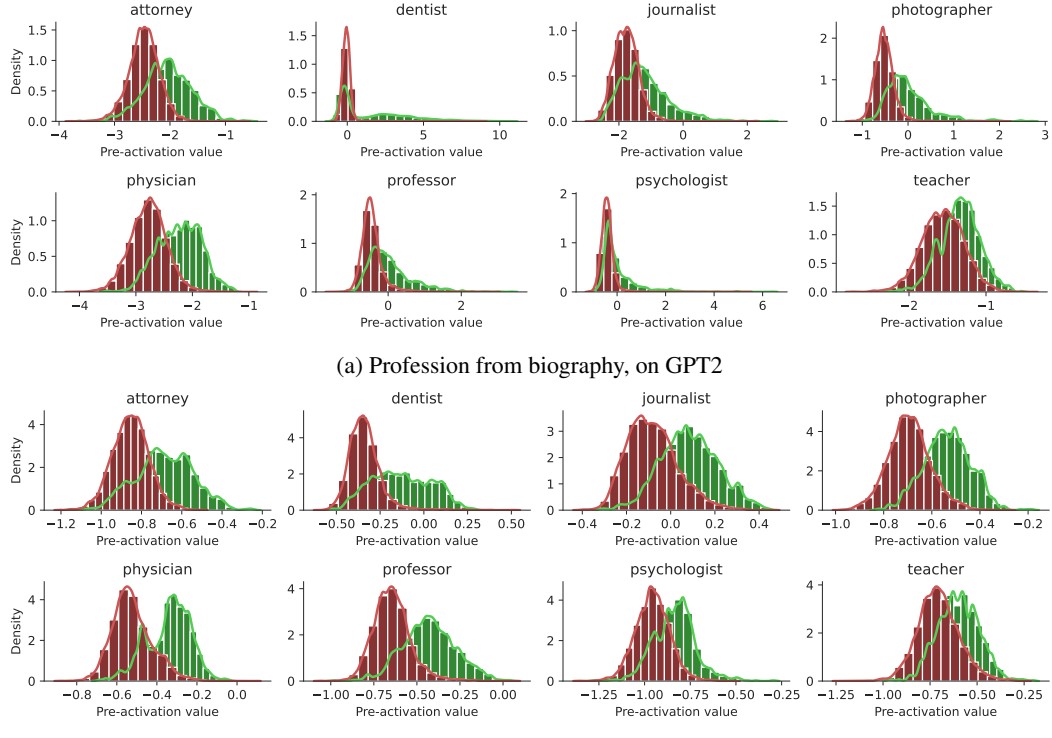

(a) Profession from biography, on GPT2

(b) Profession from biography, on Pythia-410m

Figure 14: **[2/2] Sample-level** sparse probing results on individual experts for MxDs; here we plot the values of the expert pre-activation for **positive**/**other** classes (in the 1-vs-all setting).

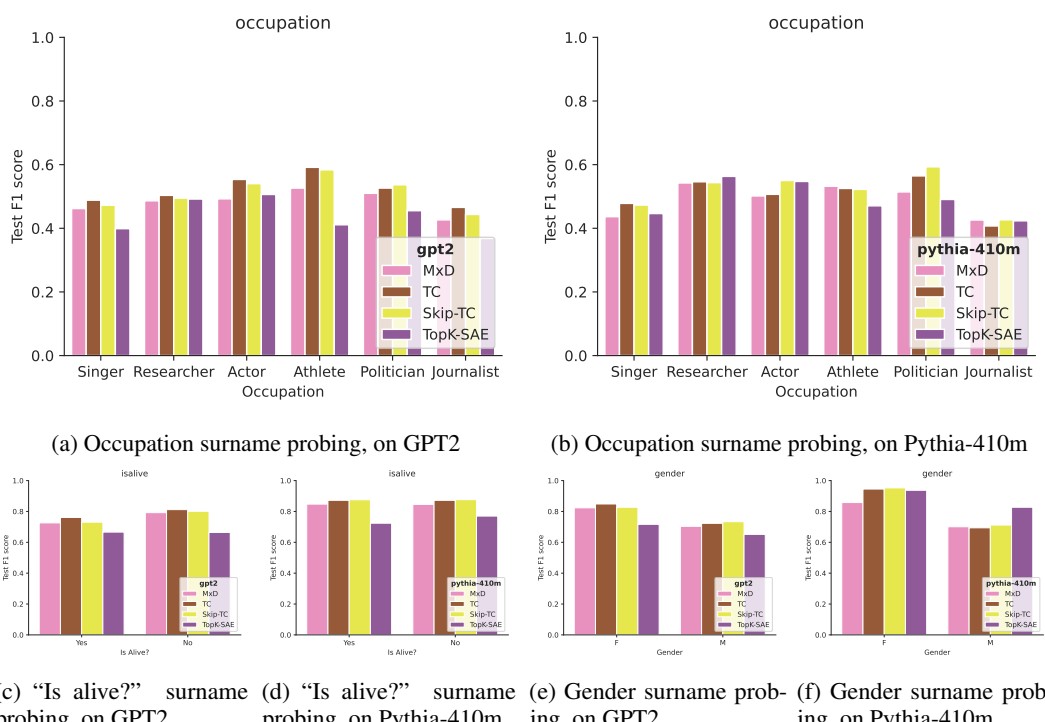

(a) Occupation surname probing, on GPT2

(b) Occupation surname probing, on Pythia-410m

(c) "Is alive?" surname probing, on GPT2

(d) "Is alive?" surname probing, on Pythia-410m

(e) Gender surname probing, on GPT2

(f) Gender surname probing, on Pythia-410m

Figure 15: **Token-level** sparse probing results on individual experts/features; the best F1 score on a held out set is presented.

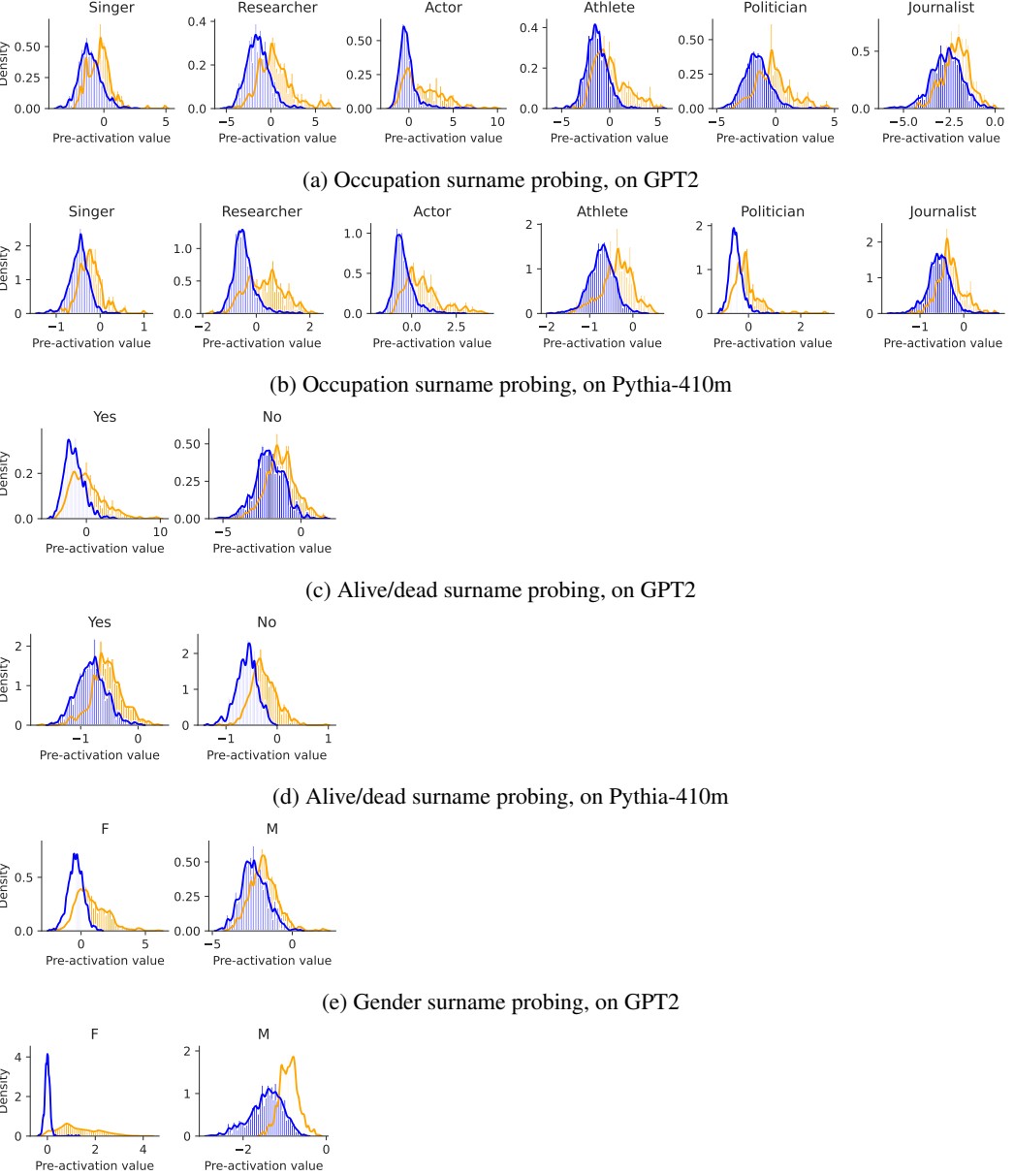

(a) Occupation surname probing, on GPT2

(b) Occupation surname probing, on Pythia-410m

(c) Alive/dead surname probing, on GPT2

(d) Alive/dead surname probing, on Pythia-410m

(e) Gender surname probing, on GPT2

(f) Gender surname probing, on Pythia-410m

Figure 16: **Token-level** sparse probing results on individual experts for MxDs; here we plot the values of the expert pre-activation for **positive**/**other** classes (in the 1-vs-all setting).

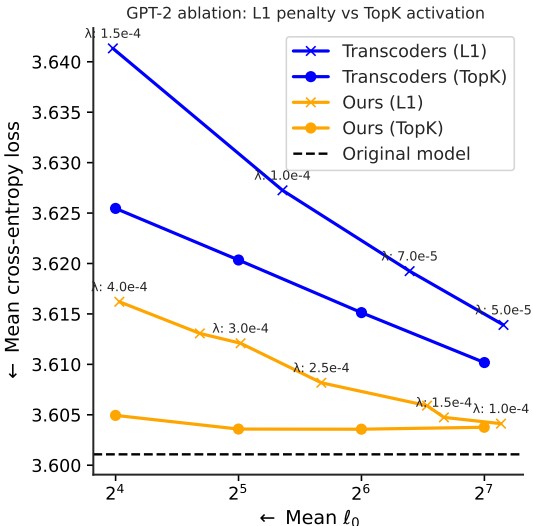

Figure 17: ReLU+TopK activation function [23] vs ReLU w/ L1 sparsity penalty [22]: both MxDs and Transcoders better recover the cross entropy loss with the TopK activation.

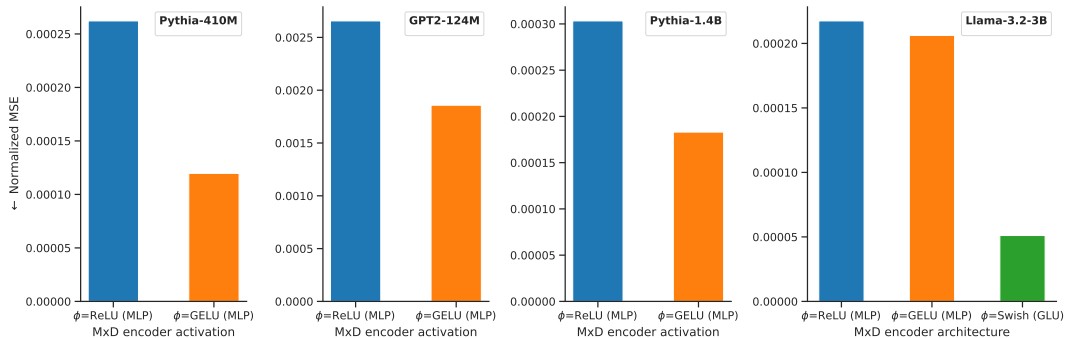

Figure 18: **Encoder architecture ablation**: MSE loss when using ReLU activation vs the GELU used by the base models; and MLPs vs GLUs for Llama (rightmost subfigure).

first 3 left-most subfigures, we see the Normalized MSE is as low as half when using GELU vs the non-native ReLU activation.

We next ablate the impact of inheriting the same encoder as the `Llama-3.2-3B` base model. In the rightmost subfigure of Figure 18, we train MxDs with ReLU-MLP, GELU-MLP, and Swish-GLU encoders. As can be seen, using a GLU with a Swish activation model (matching the base model architecture) yields a Normalized MSE almost an **order of magnitude** smaller than MLPs with GELU/ReLU.

## C  Feature balance and shared experts

### C.1  Expert/feature balance

Following the code of [27, 109], we log how often each unit of specialism/feature is used, over a fixed window of $\sim 1$M tokens. We show in Figure 19 the feature frequency at the end of training, where we observe that MxDs see a similar healthy balance of experts to the frequency of usage of features in the baselines.

Interestingly, we observe a small peak of experts that fire more frequently in MxDs (e.g., around -2 on the x-axis)–perhaps specializing in common patterns and primitives in natural language.

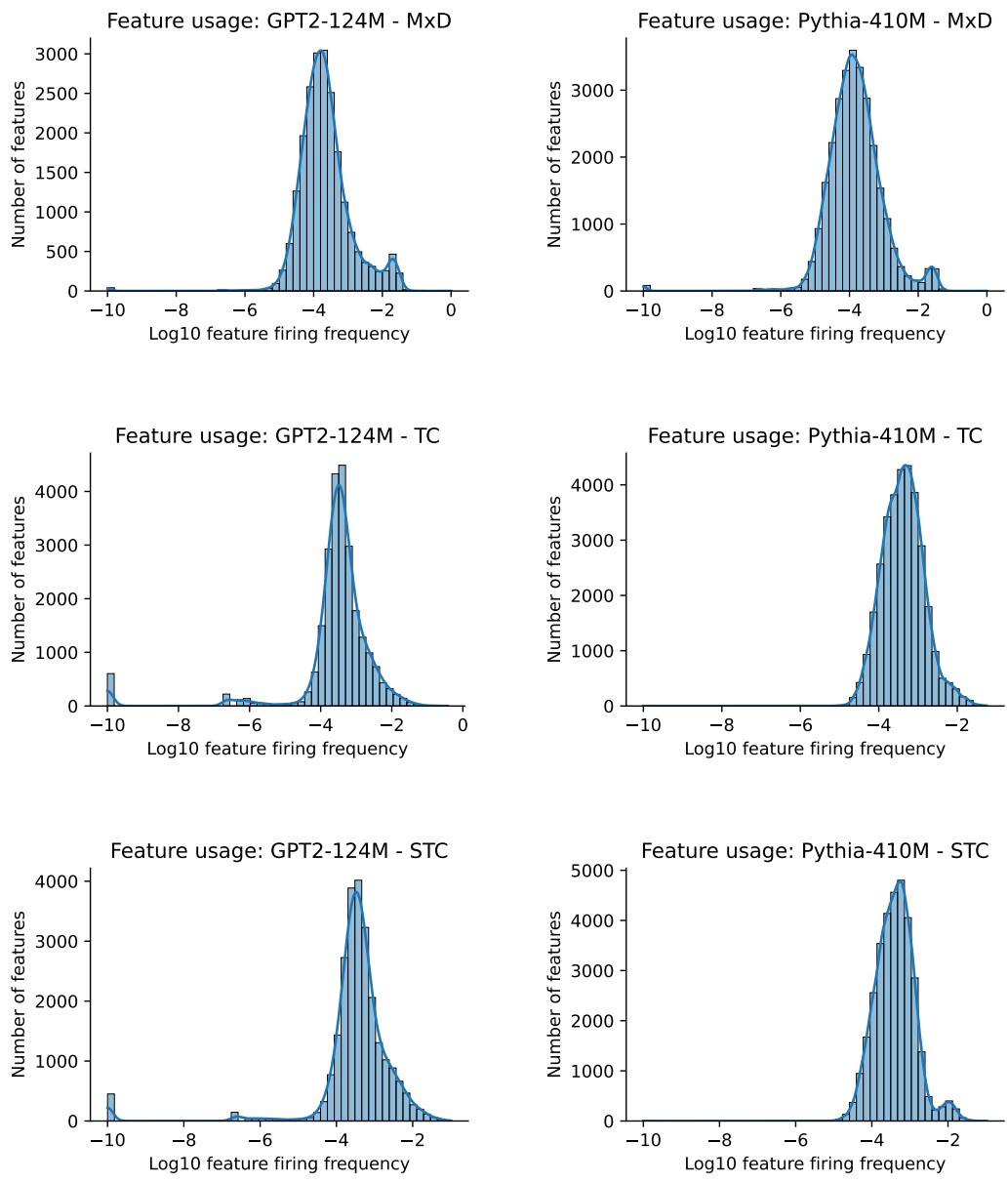

Figure 19: $\log_{10}$ feature sparsity (following [27, 109]); MxDs' experts are well-balanced, similar to the baselines' features.

## C.2 Shared experts

We find that, by default, our MxD models naturally learn to use a shared expert, with the remaining experts exhibiting strong specialization in a wide range of themes and linguistic patterns. The use of a shared expert is becoming an increasingly popular design choice, including in the latest Llama 4 models [44]–we therefore allow this pattern to emerge naturally in our base models, further justified through the evidence in [43] that shared experts can enhance specialization among the remaining experts [43]. We highlight, however, that a simple trick of sampling $\hat{K} \sim \text{Unif}\{K - K/a, K + K/a\}$ for the Top-$\hat{K}$ activation at train-time (for e.g. $a := 2$) is sufficient to remove the dominating shared-expert at minimal hit to reconstruction performance, if desired.

We train two sets of models with a base $K = 32$ on `GPT2-small` and `pythia-410m`, using $a := 2$. We first show in Figure 20 the indices of the top-activating experts for the 2 model variants on a template prompt, after training has finished. On the left-hand side of Figure 20, the models route all tokens through the same shared expert at position 1. However, we see on the right-hand side that training with the 'random-K' strategy breaks the dependence on a shared expert in position 1. Furthermore, we include in Figure 21 the corresponding train-time MSE loss for the 4 models here as ablations–observing that the random-K strategy also brings comparable performance. Based on these experiments, we recommend this simple training strategy if one desires MxD models without shared experts.

**Expert diversity/collapse**   *To what extent is the shared expert simply learning the base decoder itself?* In an initial attempt to study this, we take the same MxD trained with $K = 64$ on GPT2, and tabulate the Frobenius norm of differences between the original model's decoder $\mathbf{D}^*$, our learned base decoder $\mathbf{D}$, and the "shared expert" weight matrix $\mathbf{W}_{\text{shared}} = \mathbf{D} \text{diag}(\mathbf{c}_{\text{shared}})$ (with index shared $= 19772$), all of which are matrices of shape $(H \times O)$. As can be seen in Table 9, their difference is significant, showing that the shared expert does not simply learn the original decoder.

As discussed in the limitations section, we highlight that MxDs as formulated contain no explicit loss terms to discourage expert collapse, or encourage expert balance or diversity. Whilst we find no need for these in our experiments, they may prove useful (or even necessary) in other settings or future work.

Table 9: Distance between learned shared expert, base decoder, and original decoder

| $\|\mathbf{W}_{\text{shared}} - \mathbf{D}^*\|_F$ | $\|\mathbf{D} - \mathbf{D}^*\|_F$ | $(\|\mathbf{W}_{\text{shared}} - \mathbf{D}^*\|)_{:3,:3}$ (9 elements of the difference matrix) |
|:---:|:---:|:---:|
| 208.08 | 221.6934 | $\begin{bmatrix} 0.09 & 0.03 & 0.06 \\ 0.21 & 0.12 & 0.07 \\ 0.17 & 0.15 & 0.11 \end{bmatrix}$ |

# D   Detailed experimental setup

We list in Table 10 the resources used for each experiment: the GPU and the indicative run-time for a single model. The `mlp_expansion_factor` column refers to the expansion factor applied to the input dimension to generate the MLP width in the sparse layers (i.e. $H := I \cdot \text{mlp\_expansion\_factor}$).

Table 10: Total training time and resources used to produce the $k = 32$ experiments (the required compute being roughly the same across models trained with different $k$).

| Model | GPU used | VRAM | Training time | d_in | mlp_expansion_factor | Asset link |
|---|---|---|---|---|---|---|
| GPT2-124m | x1 GeForce RTX 3090 | 24GB | 8h 34m 37s | 768 | 32 | https://huggingface.co/docs/transformers/en/model_doc/gpt2 |
| Pythia-410m | x1 GeForce RTX 3090 | 24GB | 8h 35m 17s | 1024 | 32 | https://huggingface.co/EleutherAI/pythia-410m |
| Pythia-1.4B | x1 A100 | 80GB | 23h 25m 23s | 2048 | 32 | https://huggingface.co/EleutherAI/pythia-1.4b |
| Llama-3.2-3B | x1 A100 | 80GB | 2d 3m 50s | 3072 | 32 | https://huggingface.co/meta-llama/Llama-3.2-3B |

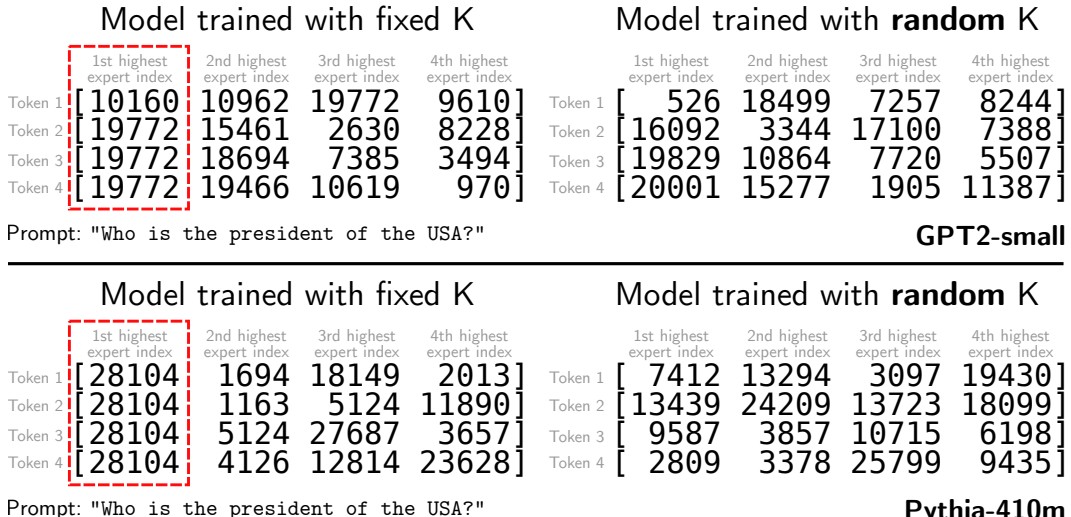

Figure 20: Top-activating experts for template prompt with and without using a randomized value of $K$ at train-time for TopK expert selection: randomization largely prevents a shared expert. Shown are the leading 4 tokens and expert indices.

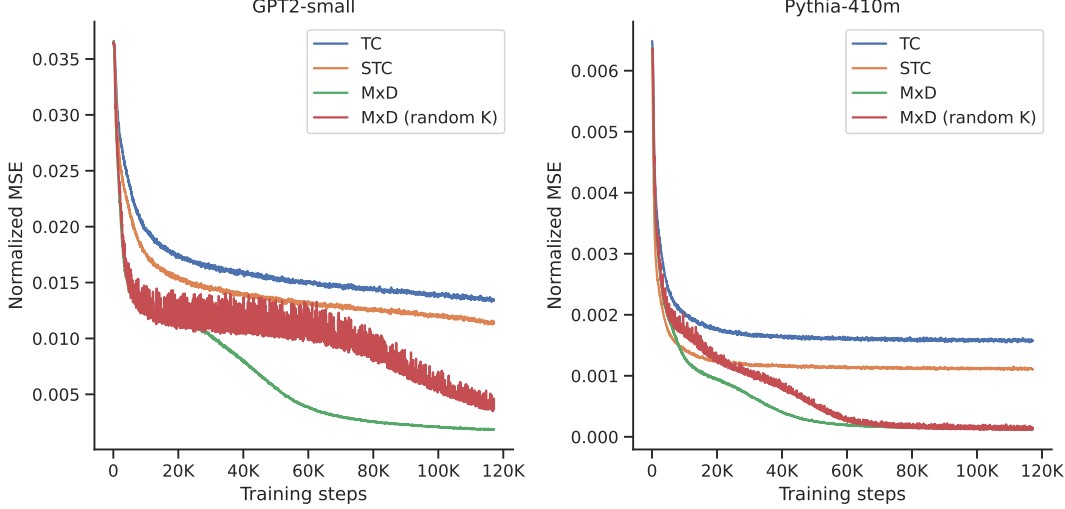

Figure 21: **MxD performance with random K sampling**: Normalized MSE loss as a function of training steps using a fixed Top $K := 32$ expert selection and when sampling $\hat{K} \sim$ Unif$\left\{K - \frac{K}{2}, K + \frac{K}{2}\right\}$.

### D.1 Feature steering details

For the steering experiments, we use two LLM judges to grade generations on two axes. The full template prompt we feed to `gemini-2.0-flash` and `llama-4-scout-17b-16e-instruct` is as follows (note that line breaks and emphases are included here only to aid visualization):

---

**Prompt given to LLM judges**

You are an expert evaluator of synthetic text.
**TASK**: Rate a collection of {num_samples} samples along two independent axes.
**AXIS 1 – CONCEPT COHERENCE:**
0.00 no shared concepts/themes/style.
0.25 faint overlap.
0.50 some overlap or similar structure.
0.75 mostly the same concepts or structure; a few partial drifts.
1.00 all snippets clearly share the same concepts, themes, style, or structure.
**AXIS 2 – GRAMMATICAL FLUENCY:**
0.00 incomprehensible.
0.25 dense errors; meaning often obscured.
0.50 frequent errors; meaning still mostly recoverable.
0.75 minor errors that rarely hinder comprehension.
1.00 completely grammatical and natural.
(Do not penalise fluency if a snippet starts or ends abruptly.).
**SCORING:** Choose any real value in [0, 1] for each axis.
**OUTPUT FORMAT:** Respond with exactly two numbers formatted '0.00, 0.00' in the order [coherence, fluency] and no other text or symbols.
**TEXT TO EVALUATE:** {samples}

---

## E  Additional qualitative results

We show in Figures 22 and 23 tokens activating the first 9 experts as they appear numerically. We sample 6 bins of expert coefficient value to show both tokens that highly activate the experts and those that do so only mildly. As can be seen, both high- and low-level specializations emerge in both GPT and Pythia models.

Whilst we observe specializations to a range of concepts (such as punctuation, MMO games, words in specific contexts), we do not notice any systemic differences between the types of expert specializations that emerge between the two models in MxD layers.

# **Pythia-410m**: Tokens routed to specific experts

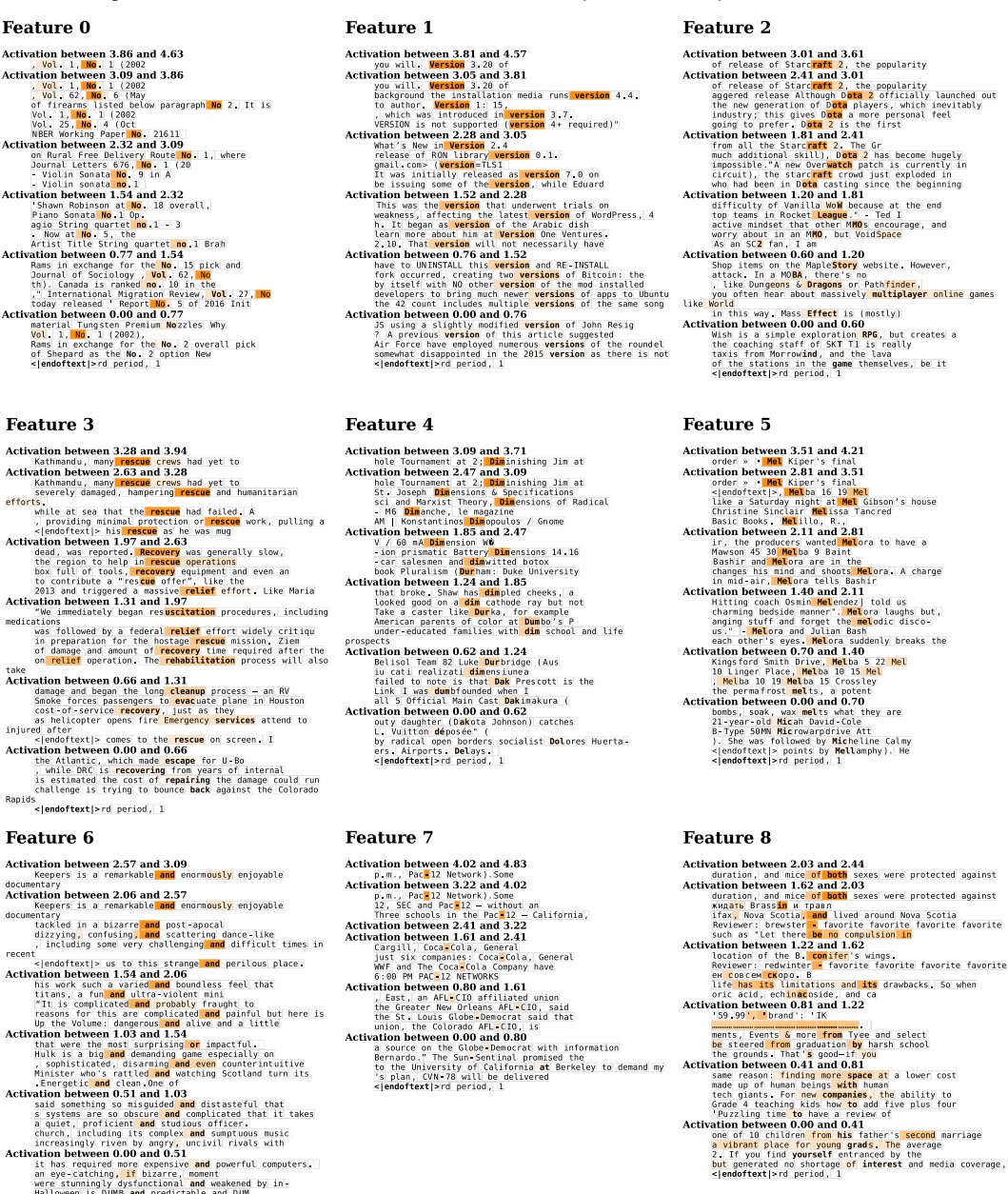

**Figure 22:** Tokens activating the first 9 numerical experts on MxDs with $K = 32$ trained on `Pythia-410m`; we sample 6 bands of activations to show both tokens that highly activate experts and those that activate them only mildly. Magnitude of activation is denoted by the **orange** highlight. Moderate specialism emerges, e.g., to MMO games, abbreviations, and words in specific contexts.

# GPT2-124m: Tokens routed to specific experts

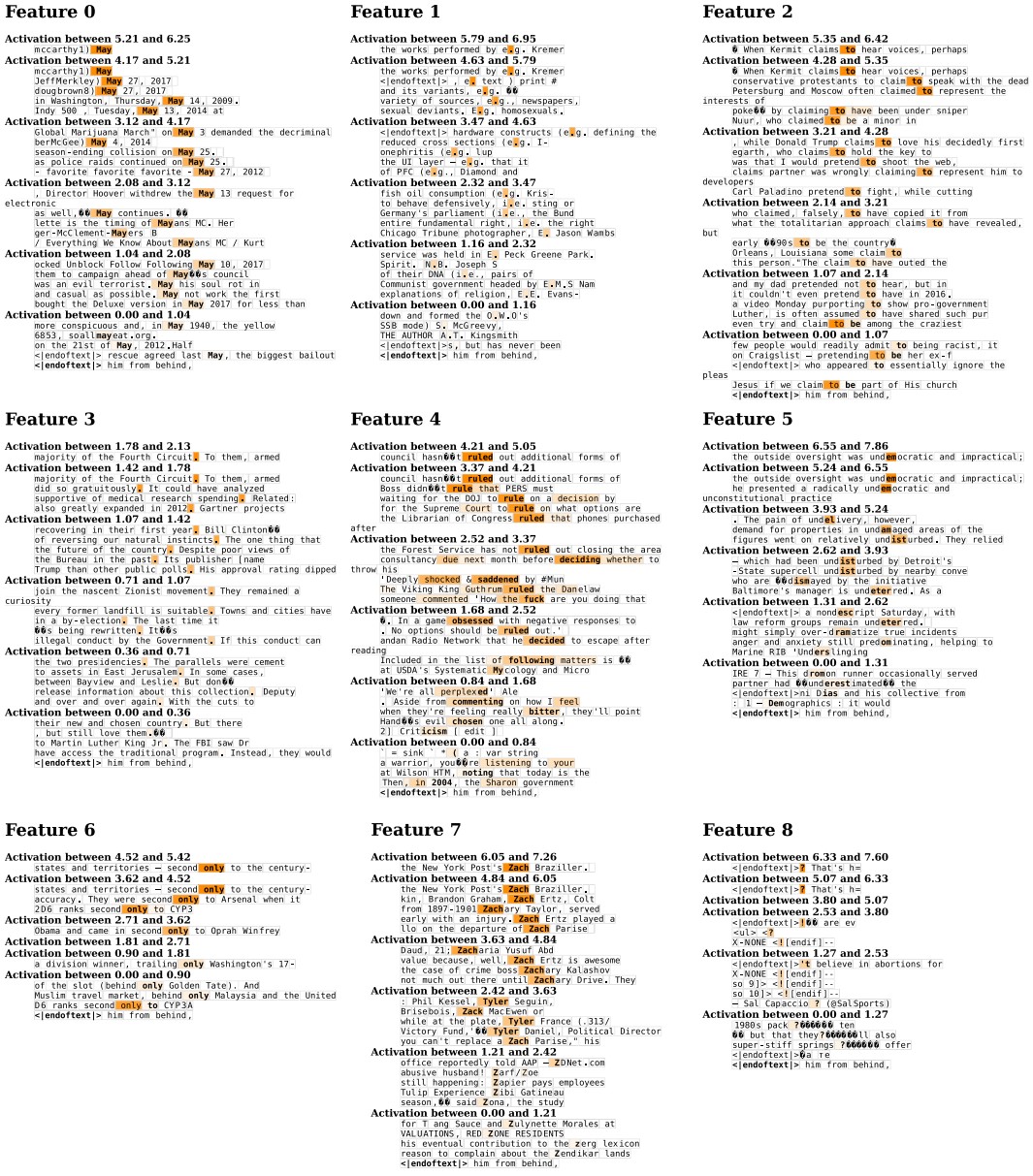

**Figure 23:** Tokens activating the first 9 numerical experts on MxDs with $K = 32$ trained on GPT2-124m; we sample 6 bands of activations to show both tokens that highly activate experts and those that activate them only mildly. Magnitude of activation is denoted by the **orange** highlight. Moderate specialism emerges, e.g., to punctuation, names, and months.