# OpenReview forum: "Towards Interpretability Without Sacrifice: Faithful Dense Layer Decomposition with Mixture of Decoders"
_NeurIPS.cc/2025/Conference — NeurIPS 2025 poster_

### Official Review · Reviewer_JDFb · 2025-07-02

**Clarity:** 3
**Significance:** 3
**Originality:** 3
**Rating:** 5
**Confidence:** 4

**Summary:**

The paper proposes the Mixture of Decoders (MxD) for post-hoc interpretation of MLPs in LLMs. The architecture achieves sparsity by approximating MLP with $N$ sparse transformations (or experts), which is unlike the existing sparse neuron approximations. The paper also proposes the Hadamard weight factorization, which allows for simplified implementation.

**Questions:**

1. Can the authors clarify how MxD can be used as a better interpretation tool by future research?
2. Can the authors clarify the role of the shared expert?

**Ethical Concerns:**

["NO or VERY MINOR ethics concerns only"]

**Final Justification:**

During the rebuttal, my major concerns (no advantage in terms of interpretability evaluation and the potential collapse to shared experts) are addressed.

**Limitations:**

Yes

**Quality:**

3

**Strengths And Weaknesses:**

Strengths:
1. The idea of sparse experts at the layer level is interesting and distinct from prior neuron-level sparsity formulations. It is nice that the paper also includes discussions about MxD's implementation and extension.
2. MxD achieves a better sparsity-faithfulness trade-off across four models with sizes ranging from 124M to 3B. It also performs stably across different sparsity levels.
3. The paper includes both faithfulness and interpretability evaluation of its approach, which is a fair experimental setup.

Weaknesses:
1. MxD does not show advantages over baselines on interpretability, based on the experimental results of probing and steering. Since the ultimate goal of this line of research is to enable better interpretation of MLPs, this approach does not particularly provide an advancement. On the other hand, the observation that MxD achieves higher faithfulness than existing approaches but only comparable interpretability also points out its technical weakness, i.e., its loss function does not particularly advocate better interpretability.
2. The role of the "shared expert" in enabling the faithfulness gain needs more clarification. A key claim of the paper is that MxD improves reconstruction faithfulness under high sparsity via expert specialization. However, the presence of an emergent shared expert makes it unclear whether the gain stems from true specialization or this general-purpose expert. In an extreme case, it could happen that this shared expert replicates most of the original MLP and thus presents high faithfulness. However, relevant to Weakness 1, there's no particular architecture or loss function design that prevents this situation.
3. Presentation: while the overall paper writing is clear, I find Fig 1 particularly hard to understand (or not intuitive enough without reading the technical details in the main text). For the neuron-level sparsity figure, the notation of $N$ corresponds to $H$ in Eq (2). It may be better to use the notations more consistently across the paper.

---

> ### Author Rebuttal · Authors · 2025-07-30
>
> We are grateful to the Reviewer JDFb for praising the idea to move to layer-level sparsity as interesting, and highlighting the stability of MxDs to varying levels of sparsity. We address each of the raised concerns and questions below:
>
>
> ## [W1] Advantages for interpretability
>
>
> > ”Since the ultimate goal of this line of research is to enable better interpretation of MLPs, this approach does not particularly provide an advancement”
>
> We respectfully disagree with the reviewer--throughout the abstract [L6] and introduction [L56:L65], we argue that the benefit of MxDs lies in overcoming the accuracy trade-off plaguing sparse layers. MxDs consequently *do* advance interpretability, by removing the large cost to model performance incurred when adopting sparse layers in place of original model components [1]. **Two reviewers highlight this key contribution**:
>
>
> - **Reviewer-zUPk** praises that MxDs `help(s) address a very core issue in interpretability work where sparse approximations greatly underperform`, constituting `a potentially viable replacement for the original component.`
> - **Reviewer-9H1M:** `tackles the core trade-off between performance and interpretability` and `central challenge in AI safety [...] making the research highly relevant and important`
>
> Would any additional experiment or metric convince the reviewer about the value of the proposed method? We are open to additional experiments if the reviewer has a specific hypothesis to test.
>
>
> ## [W2 & Q2] Shared experts
>
>
> We respectfully clarify that `"A key claim of the paper is that MxD improves reconstruction faithfulness under high sparsity via expert specialization"` is **not** the claim we make in our submission. In [L76, L145] we attribute the reconstruction faithfulness specifically to the *full-rankness* of the weights that MxD provides (independent of specialization).
>
> Please see Sect. C.2 and Fig. 21-22 in the appendix, where we show that **a single shared expert can be avoided if desired by training with a randomized top-K selection**, at very small cost to reconstruction loss. For the same GPT2 $K=64$ model, we tabulate below the Frobenius norms of differences between the original model’s decoder $\mathbf{D}^*$, our learnt base decoder $\mathbf{D}$, and the “shared expert” weight matrix $\mathbf{W}_\text{shared}=\mathbf{D}\text{diag}\(\mathbf{c}\_{\text{shared}})$ (with index $\text{shared}=19772$), all of which are matrices of shape $H\times O$. As can be seen, their difference is significant, showing that the shared expert does not simply learn the original decoder:
>
> | $\vert\vert \mathbf{W}_{\text{shared}} - \mathbf{D}^* \vert\vert_F$ | $\vert\vert \mathbf{D} - \mathbf{D}^* \vert\vert_F$ | $\left(\vert \mathbf{W}_{\text{shared}} - \mathbf{D}^* \vert\right)\_{:3,:3}$ (9 elements of the difference matrix) |
> | --------------------------------------------------------------------- | ----------------------------------------------------- | ----------------------------------------------------------------------------------------------------------------------- |
> | 208.08                                                                | 221.6934                                              | [[0.09, 0.03, 0.06],  [0.21, 0.12, 0.07],  [0.17, 0.15, 0.11]]                                                    |
>
> We are grateful to the reviewer for the chance to address this further, however--we indeed use no explicit loss discouraging degenerate solutions, finding it sufficient to rely on the random weight initialization alone. We will expand on our discussion of this on [L327] in the limitations as follows:
>
>
>     "Secondly, MoEs are prone to issues of expert imbalance [71] or collapse [88]. Through random weight initialiazation alone, we find MxDs in our experiments learn diverse experts without collapsing to the base decoder. However, we expect standard MoE load-balancing [2] or diversity losses [3] to be useful for MxDs should one need more explicit ways of encouraging expert diversity, or when training new interpretable MxD architectures end-to-end."
>
>
> ## [W3] Fig. 1 presentation
>
> We appreciate the constructive feedback about Figure 1--we will indeed use $H$ in place of $N$ to denote the number of specialized units in the centre subfigure in the revised manuscript. Would the reviewer have any more specific suggestions on how to improve readability?
>
> ## [Q1] Can the authors clarify how MxD can be used as a better interpretation tool by future research?
>
> By showing the feasibility of alternative layer designs that largely overcome the sparsity-accuracy trade-off, we argued that MxDs are an important step towards interpretable layers capable of replacing dense MLPs of base models, distilling specialized computations directly into the new model. One immediate application for future research that we are excited about is **always-on feature monitoring**. With the updated base model computation consisting of tens of thousands of specialized experts, safety-relevant features could be tracked, ablated, and manually tweaked at inference time without any additional post-hoc cost.
>
> Secondly, MxDs provide more promising evidence that the dominating Mixture of Experts (MoE) paradigm [4] can yield interpretable subcomputations [5,6,7]. Given that a large number of new foundation models are MoE models [8,9], we hope the study in the MxD paper might advocate for the research community to focus its efforts on interpretable units of computation in existing foundation models, and/or design new interpretable-by-design architectures for training end-to-end.
>
>
> ----------
> - [1]: Elhage, et al., "Softmax Linear Units", Transformer Circuits Thread, 2022.
> - [2]: Wang, Lean, et al. "Auxiliary-loss-free load balancing strategy for mixture-of-experts." *arXiv preprint arXiv:2408.15664* (2024).
> - [3]: Liu, Boan, et al. "Diversifying the mixture-of-experts representation for language models with orthogonal optimizer." *arXiv preprint arXiv:2310.09762* (2023).
> - [4]: Shazeer, Noam, et al. "Outrageously large neural networks: The sparsely-gated mixture-of-experts layer." *arXiv preprint arXiv:1701.06538* (2017).
> - [5]: Park, Jungwoo, et al. "Monet: Mixture of monosemantic experts for transformers." *ICLR 2025.*
> - [6]: Oldfield, James, et al. "Multilinear mixture of experts: Scalable expert specialization through factorization." *NeurIPS 2024.*
> - [7]: Yang, Xingyi, et al. "Mixture of experts made intrinsically interpretable." *ICML 2025.*
> - [8]: Meta AI. “The Llama 4 herd: The beginning of a new era of natively multimodal AI innovation.” 2025.
> - [9]: Guo, Daya, et al. "Deepseek-r1: Incentivizing reasoning capability in LLMs via reinforcement learning." arXiv preprint arXiv:2501.12948 (2025).
>
> ---
>
> We welcome **Reviewer JDFb** to ask any additional questions that might raise their support of the paper--we will gladly answer them.

---

> > ### Comment · Reviewer_JDFb · 2025-08-06
> >
> > Thank you for your response!
> >
> > W1 and Q1: I'd like to clarify that, by "interpretability", I didn't refer to the general research field of interpretability. Instead, I mainly referred to the interpretability evaluation of MxD in Sec 3.2, based on the experimental results of probing and steering. My major concern is that MxD does not show an advantage over baselines in the probing and steering experiments, which implies that, despite the better sparsity-faithfulness trade-off it achieves, it does not seem to advance practical utility (e.g., enabling significantly more effective probing or steering). Could the authors further respond to my concern?
> > I acknowledge the use case of "always-on feature monitoring" mentioned by the authors, but this can be similarly achieved by the baseline approaches as well. For MoE, I agree with the authors on MxD's unique contribution along the line.
> >
> >  W2 and Q2: My concern is addressed. It will help if the authors can add the discussion to clarify the shared expert observation.
> >
> > W3: As long as the notations are consistent between the figure and the main text, it is good!

---

> > > ### Author Response · Authors · 2025-08-06
> > >
> > > Dear reviewer,
> > >
> > > Thank you for the response--we are pleased to hear that your original concerns are addressed (aside from the below), and we are grateful for your willingness to engage in discussion!
> > >
> > > ## Remaining concern: [W1+Q1]
> > >
> > > The reviewer is correct to note that MxDs do not outperform the baselines in probing and steering benchmarks--as described on e.g. [L180], [L224], the goal of these experiments is to show that **MxDs retain all the interpretability benefits of existing techniques whilst also being much more accurate layer approximators**. One of our core arguments in the introduction is that layer faithfulness in MxDs is an important and practically useful property in its own right:
> > >
> > > Whilst the baselines (e.g., Transcoders) can indeed be used for monitoring of select features, their failure to accurately replicate the original MLPs means they inevitably run in addition to the original layers, post-hoc.
> > >
> > > In contrast, MxDs’ accurate approximations take a big step towards being able to swap out the dense MLPs in the base model with a computation that is intrinsically interpretable. Faithful layers like MxDs enable the possibility of a drop-in component that would provide **full control and transparency over the computation**--all N of the subcomputations that directly make up the output are **monitored by construction in every forward pass** (this is the notion of “always-on” we intended to refer to in our rebuttal), and steerable by simply increasing the value of an expert neuron.
> > >
> > > Additionally, we envision MxD-style subcomputation in future research might also simplify removal of harmful concepts; rather than needing to remove an SAE/Transcoder feature direction from an entangled MLP output vector, one can exclude a problematic MxD expert from ever appearing in the computation in the first place.
> > >
> > > We believe that these properties demonstrate MxD faithfulness as a practical interpretability advance that the community would be interested in (bridging both post-hoc and intrinsically interpretable methodologies), and we hope this response alleviates the final concern of the reviewer.

---

> > > > ### Comment · Reviewer_JDFb · 2025-08-08
> > > >
> > > > Thank you for your further clarification! This is very helpful. I hope you can include them in the next version. I don't have further concerns at this moment. I will increase the rating accordingly.

---

> > > > > ### Author Response · Authors · 2025-08-08
> > > > >
> > > > > Dear Reviewer JDFb,
> > > > >
> > > > > Thank you for your time and effort in engaging in the discussion! We are pleased that our response addressed your remaining concerns, and we appreciate your raising the rating accordingly.
> > > > >
> > > > > We will include these discussions in the revised manuscript--if you later think of any ways we could further improve the paper's clarity during the AC-reviewer discussion period, please do not hesitate to let us know.
> > > > >
> > > > > The authors.

---

### Official Review · Reviewer_8ypg · 2025-07-02

**Clarity:** 3
**Significance:** 3
**Originality:** 3
**Rating:** 5
**Confidence:** 4

**Summary:**

The paper proposes an inherently interpretable method that complements existing neuro-sparsity based methods, e.g. sparse overcomplete autoencoders (SAEs) and Transcoders, by imposing sparsity constrains at the layer level.

More specifically, proposed Mixtture of Decoders (MxD) method, replaces MLP components present in standard LLMs by an autoencoder where the decoder part is defined as a sparse mixture of decoders who match in number of parameters/capacity the original MLP it aims to replace. In doing so the proposed method is capable of learning specialized expert (decoder) branches that are interpretable.

Experiments on 4 LLMs (~3B parameters), i.e. GPT2-124M [3], Pythia-410m, 187 Pythia-1.4b [41], and Llama-3.2-3B [1], show that the proposed method are comparable with the state of the art methods (SAEs and Transcoders) while having a performance that is significantly robust to varying levels of sparsity.

**Questions:**

- As indicated in the manuscript the proposed methods introduces the capability of learning a representation among a mixture of sparse decoders. In this regard the training process of the proposed method is significantly more complex that of the considered competitors (SAEs and TCs). May you provide some details on how the training processes of these methods compare w.r.t. each other?

- In L. 196-198, it i stated that there is the expectation that the proposed MxDs are capable of scaling to significantly larger models. May you provide some theoretical insights or perhaps an intuition of why this would be the case? The manuscript would benefit from having this pointer further

    elaborated.

- In l.209-210, it is stated that [17,19] find sparse solutions to be more interpretable. Are there any guarantees, or perhaps clear indicators, that the findings from those articles are transferable to the proposed setting?

- In Sec. 3.2.2, two LLMs are used as gold standard for measuring properties of the output produced by feature steering. Since these are models that are not perfect, is there is pointer to existing work showing the validity of this practice/protocol?

**Ethical Concerns:**

["NO or VERY MINOR ethics concerns only"]

**Final Justification:**

The clarification on the faithfulness loss effectively clarified my initial misunderstanding w.r.t. [W1 & L3].

The suggested Revised Limitations text and other actions indicated to be taken to address [W2 & Q4 & L5] seem adequate. Likewise for my question (Q3) regarding the relevance of [17,19], i.e. the suggested course of action seems transparent enough and adequate.

Regarding [W3 & L4], thanks for the indicated modification (of moving fig. 10 into the manuscript), I believe this would be  good complement to the results reported on predictive performance (fig.3). I would encourage the authors to include the pointers on the training complexity (Q1), present in the rebuttal, in the main body of the paper. Beyond that, I  understand it can be challenging to present complex and/or extensively-validated ideas in the given page limits. However, that is a constraint we need to deal with and for which a proper trade-off must be found.  Sometimes, aspects like this make the difference between two papers with two good ideas.

Regarding [L1 & L2], positioning with respect to related work on sparse decomposition methods, I acknowledge two of the references are very recent. While I do not consider the  missing positioning with respect to those two as grounds for rejection, I would still encourage the authors to include them in their related work section for completeness. The work from Pearce et al., is a bit more older, I would be more strict on the inclusion of that one.

**Limitations:**

In addition to the points listed as weaknesses I would stress the following aspects:

- In the related work section (Sec. 4), the aspect related to Sparse Decomposition, i.e. the core direction of this paper, limits itself to listing some existing efforts. This section would benefit with an explicit positioning on how the proposed method compares w.r.t. existing efforts.

    In addition, The proposed method seems to bear resemblance to the work from [Pearce et al, 2025], [Dooms et al., 2025] [Adler et al., 2025] (please see below for complete references). These works do not seem to be discussed in the manuscript.

- One of the metrics used for the evaluation of the proposed method is “faithfulness” to the base model composed by standard MLPs. In this regard, the evaluation seems to be equating comparable predictive performance with faithfulness. While there should be the requirement to meet (or improve) the performance already achieved by the original model, meeting this requirement does not guarantee that new model will learn the same representation as that learned by the original.

- Empirical evidence, e.g reconstruction result (Fig 10), output space similarity (Sec. B.1), etc., used to support statements made in the paper are actually  delegated to the Appendix. This gives the impression of the paper not being self-contained. A better balance needs to be achieved here as to ensure statements made are paired with proper evidence (or pointers to the literature).

- In Sec. 3.2.2, two LLMs are used as gold standard for measuring properties of the output produced by feature steering. Since these are models that are not perfect, this casts doubts on the observations made from this results.


**References**

- Pearce et al. "Bilinear MLPs Enable Weight-based Mechanistic Interpretability" ICLR 2025.

- Dooms et al. "Compositionality Unlocks Deep Interpretable Models" AAAI 2025 ColoRAI Workshop

- Adler et al, "Towards Combinatorial Interpretability of Neural Computation" arXiv:2504.08842

**Paper Formatting Concerns:**

N.A.

**Quality:**

3

**Strengths And Weaknesses:**

**Strengths**

- To a good extent the manuscript has a very clear structure and the way content is presented, to a good extent, provides a very good and clear flow in the presentation of content.

- The proposed method seems to provide comparable if not superior performance to that of state of the art methods.

- The theoretical proofs put in place show, the proposed method has the capability to provide the desired sparsity properties without sacrificing information in the process, e.g. as is the case with SAEs which lose some information due to imperfect reconstruction processes.

- Several resources are released in order to ease the testing of the proposed method.


**Weaknesses**

- The proposed evaluation seems to equate equal predictive performance to “faithfulness” to the original (standard MLP) model.

- Some of the adopted evaluation procedures (model faithfulness and feature steering) do not seem to be adequate.

- Some relevant parts of the manuscript have been delegated to the appendix.

---

> ### Author Rebuttal · Authors · 2025-07-30
>
> We are grateful to **Reviewer-8ypg** for acknowledging the robustness of MxDs to varying levels of sparsity and the strengths of the theoretical results provided. We address each of the concerns below:
>
> ## [W1 & L3] Predictive performance vs faithfulness
>
> We inherit the term “faithfulness” from prior work [1], which uses the term synonymously with predictive ability, e.g.:
>
>     "which seek to faithfully approximate a densely activating MLP layer with a wider, sparsely-activating MLP layer"
>
> Furthermore, the reconstruction loss in [1] is named a “faithfulness loss”. We respectfully disagree with the reviewer's claim in the limitations that `meeting this requirement does not guarantee that new model will learn the same representation as that learned by the original`: through the reconstruction loss, the layer mapping is indeed trained to reconstruct the same MLP output vector $\mathbb{y}=\text{MLP}(\mathbf{x})\in\mathbb{R}^O$  as closely as possible; we are not training on the final downstream cross-entropy loss.
>
> We are open to specific suggestions that the reviewer might have for how to refer to this property instead if they strongly believe in this enough to depart from the terminology of prior work.
>
> ## [W2 & Q4 & L5] Adopted evaluations & LLM as a judge
>
> We acknowledge that no evaluation in the paper is strong evidence *in isolation*. However, each experiment is designed to complement the other--probing evaluates features for which we have labels (frequently used to evaluate prior work [2,3]), whilst evaluations of steering with judge LLMs allow for a scalable evaluation of features for which we do not have labels. Taken as a whole, we argue that the experiments do substantiate the claims we make in the paper; that `"MxDs learn similarly specialized features"`  on [L15].
>
> We highlight that using **LLM-as-a-judge is the standard evaluation of steering in a recent benchmarking suite AxBench** [6] and in past work [7]. We will cite these works in the feature steering section to better ground our evaluations in standard protocol, and we will acknowledge their potential limitations in the conclusion:
>
> **Revised limitations**
> [L330]
>
>     With regards to feature evaluations, our steering experiments rely on LLMs as judges. Whilst this is commonplace in recent popular steering benchmarks [6], there is mixed evidence emerging about the reliability of all base models [4,5]. Our experiments attempt to circumvent issues by reporting scores across two capable SOTA models, but we caution inferring too much into the absolute values of reported scores (rather than the relative performance between sparse layers).
>
>
> ## [W3 & L4] Appendix results
>
> Many additional results are delegated to the appendix due to the page limit. However, we note that the results in Fig. 10 specifically referenced by the reviewer showcase the *same* experiments as those in Fig. 3 (with cross-entropy loss used in [1,8]) in the main paper, only with an alternative metric (reconstruction loss), and thus are supplementary.
>
> With the additional page available in the camera-ready revision, we will include results from Sect. B.1 in the main paper.
>
> ## [Q1] Training complexity
>
> We highlight that the forward pass cost of the proposed MxD layer is not significantly more complex--when parameter-matched to sparse MLP layers as in our experiments, each MxD forward pass has only an additional $O/2$ FLOPs.
>
> **Parameter count & FLOPs**
> We first tabulate the theoretical parameter counts and FLOPs for MxDs vs Sparse MLPs below. To be consistent with the popular PyTorch library fvcore [9] we count one fused multiply-add as one FLOP, and the Hadamard product between two $d$-dimensional vectors as requiring $d/2$ FLOPs:
>
> |                | Parameter count         | FLOPs                         |
> | -------------- | ----------------------- | ----------------------------- |
> | **Sparse MLP** | $H (I+O)$             | $H (I+O)$                   |
> | **MxD**        | $N (I+O) + H^* (I+O)$ | $N (I+O) + H^* (I+O) + O/2$ |
>
> > $I,O$ denote the input and output dimensions respectively, $H^*$ the original models’ hidden layer, $N$ the MxD expert count, and $H$ the sparse MLPs' width.
>
> **Empirical benchmarks**
> We next run benchmarks for a Sparse MLP layer vs MxD with a batch size of 512, and dimensions: $I=H^*=O=1024$, and number of experts/features as $N=8192, H=9216$.
>
> |                | Peak memory usage (MiB) | Latency (ms) | Parameter count | Reported FLOPs per sample (fvcore [1]) |
> | ---| --- | --- | --- | --- |
> | **Sparse MLP** | 386.50                  | 1.394        | 18,874,368         | 18,874,368                     |
> | **MxD**        | 389.50                  | 1.457        | 18,874,368         | 18,874,880                     |
>
> ## [Q2] Scalability to larger models
>
> We argue that the computational benchmarks above provide evidence that MxDs will be *no more difficult to scale than sparse MLP layers*, with very similar FLOPs and latency costs. For initial empirical evidence of further scalability, we perform **three additional large-scale experiments on the 27B Gemma2 model** below, where we see MxDs continue to outperform on the sparsity-accuracy frontier:
>
> | **Step**                                  | **10k**   | **20k**   | **30k**   | **40k**   | **50k**   |
> | --- | --- | --- | --- | --- | --- |
> | **Transcoder** Normalized MSE (↓)      | 0.147     | 0.132     | 0.128     | 0.123     | 0.119     |
> | **Skip Transcoder** Normalized MSE (↓) | 0.128     | 0.107     | 0.102     | 0.100     | 0.093     |
> | **MxD** Normalized MSE (↓)             | **0.098** | **0.096** | **0.093** | **0.086** | **0.069** |
>
> > **Gemma2-27B, Layer 20, K=32, 50k iterations.**
> > To fit in memory, the model is loaded in half precision, and we use 1/8th the batch size and # buffers stored, and 1/2 the multiplier on the # of latent features (with values of 4, 16, and 16, respectively).
>
> We further note that the MoE architecture itself [4] has been scaled successfully to hundreds of billions of parameters in the pre-training regime [10,11].
>
> **Revised claims**
> We will revise the claims in our discussions of scalability as follows (based on the computational benchmarks performed), to highlight that we expect MxDs to be *no more difficult to scale than sparse MLPs*:
>
> [L196]:
>
>     [...] we expect MxDs to scale just as well *as sparse MLPs* to models with tens of billions of parameters or more.
>
> [L321]:
>
>     [...] Whilst we fully anticipate this trend to *scale just as well as with sparse MLPs* in even larger models, our experiments only provide direct evidence for LLMs with up to 3B parameters, given our limited resources.
>
>
> ## [Q3] Relevance of citations [17,19]
>
> We agree that these studies are only high-level motivation: [17] shows humans prefer explanations with $\leq 32$ concepts, suggesting smaller K may aid interpretability, while [19] provides complementary quantitative metrics. To avoid over-reliance on these priors, we'll remove their appeal from Sect. 3, citing them briefly in the introduction alone.
>
>
> ## [L1 & L2] Additional related work
>
> We are grateful to the reviewer for pointing out interesting related work [12, 13, 14]; the latter two of which appearing on ArXiv a month before the NeurIPS submission deadline. We will include a discussion of how they relate to the proposed method in the revised manuscript in addition to acknowledging alternatives to interpretability-via-sparsity.
>
> **Concretely, a new section following the discussion of sparse decompositions will read:**
>
>     Interpretable layers: Whilst MxDs follow the paradigm of specialization-via-sparsity, a number of promising interpretable mechanisms have been recently proposed that do not rely on sparsity constraints. For example, recent work explore bilinear layers [12], compositional structure [13], and/or tensor networks [14] as alternative ways of designing interpretable architectures. Notably, [12] show how the Hadamard product between two linear transformations of the same input vector allows direct interpretation (such as analyzing input-output feature interactions). Whilst MxDs’ yields a similar Hadamard product forward pass, it involves two different input vectors, and establishes a theoretical equivalence to mixture of experts and conditional computation when sparsity is present in one of the operands (i.e., the expert coefficients).
>
> ---
>
> **References**
>
> - [1]: Dunefsky et al. "Transcoders find interpretable LLM feature circuits." NeurIPS 2024*.*
> - [2]: Gao et al. "Scaling and evaluating sparse autoencoders." ICLR 2025*.*
> - [3]: Gurnee, et al. "Finding neurons in a haystack: Case studies with sparse probing." *TMLR 2023.*
> - [4]: Zheng et al. "Judging llm-as-a-judge with mt-bench and chatbot arena." NeurIPS 2023.
> - [5]: Bavaresco et al. "Llms instead of human judges? a large scale empirical study across 20 NLP evaluation tasks." ACL 2025.
> - [6]: Wu et al. "Axbench: Steering LLMs? Even simple baselines outperform sparse autoencoders." *ICML 2025.*
> - [7]: Cao et al. "Personalized steering of large language models: Versatile steering vectors through bi-directional preference optimization." NeurIPS 2024.
> - [8]: Gonçalo et al. "Transcoders beat sparse autoencoders for interpretability." arXiv 2025.
> - [9]: fvcore (FAIR), detectron2 library.
> - [10]: Meta AI. “The Llama 4 herd: The beginning of a new era of natively multimodal AI innovation.” 2025.
> - [11]: Guo et al. "Deepseek-r1: Incentivizing reasoning capability in LLMs via reinforcement lerning." arXiv 2025.
> - [12]: Pearce et al. "Bilinear MLPs Enable Weight-based Mechanistic Interpretability" ICLR 2025.
> - [13]: Dooms et al. "Compositionality Unlocks Deep Interpretable Models" AAAI 2025 ColoRAI Workshop
> - [14]: Adler et al. "Towards Combinatorial Interpretability of Neural Computation" arXiv 2025.
>
> ---
>
> Might **Reviewer 8ypg** have any additional questions? If all concerns were addressed, we would otherwise be grateful if the reviewer raised their score.

---

> > ### Comment · Reviewer_8ypg · 2025-08-05
> >
> > I thank the authors for the provided responses to my original review which has clarified my initial doubts and misunderstanding ([W1 & L3]). In addition, I appreciate the clear pointers on how the manuscript would be revised accordingly.
> > I will take this and the rebuttal given to the other reviewers when making my final recommendation

---

> > > ### Author Response · Authors · 2025-08-05
> > >
> > > Dear **Reviewer 8ypg**,
> > >
> > > We are grateful for your response, and we are glad to hear the rebuttal clarified your initial doubts!
> > >
> > > Whilst the reviewer-author discussion phase remains open, might we ask if you have any follow-up questions about the paper that we could answer?
> > >
> > > Thanks kindly,
> > >
> > > The authors.

---

### Official Review · Reviewer_zUPk · 2025-07-03

**Clarity:** 3
**Significance:** 4
**Originality:** 3
**Rating:** 5
**Confidence:** 3

**Summary:**

This paper proposes to learn an interpretable approximation of LLMs via sparse learning, but at the layer level rather than previously studied neuron level. The method replaces the computation in Transformer MLPs with a k-sparse sum of linear transformations on the hidden intermediate representation, and uses a Hadamard-product based parameterization to reduce the footprint of this sparse replacement. Through a special hadamard product trick, the mixture of linear transformations becomes much more computationally feasible, and experimental results show much closer performance to baseline models, while preserving interpretability capabilities.

**Questions:**

1. Can you provide some high level justification for how we can think of the equivalence between a single expert in MxD and a single neuron in SAE as the core unit of decomposition?
2. Despite efficiency of the sparse method not being the core importance in interpretability work, and despite the Hadamard trick improving feasibility, can you comment on the efficiency of this method versus SAE and transcoders?
3. Lines 197-198 suggest that this work would scale well to larger models, but what is the justification for this claim based on the MxD construction?
4. How are sparse layers trained? Is it via layer replacement and then training just the layers? Some training details seem glossed over perhaps as a result of being in line with prior work, but these details are important for reproducibility.
5. The factorized forward pass in equation 5 is critical to understand how equations 3 and 4 relate to each other, but the rank preservation text appears between equations 4 and 5. I might suggest reordering sections 2.4 and 2.3 for reader understanding, but also additional clarity while introducing the parameterization could help.

**Ethical Concerns:**

["NO or VERY MINOR ethics concerns only"]

**Final Justification:**

The current work is clearly written, uses extensive experimental evidence to support their claims, and makes a measured contribution within the AI safety/interpretability subfield that I believe others within this field will find useful.

**Limitations:**

yes

**Quality:**

4

**Strengths And Weaknesses:**

Strengths
1. This paper presents clear experimental evidence for its claims, and presents numerous additional experiments for completeness in the appendix, like removing the shared expert or finding dead experts.
2. The utility of the induced sparsity is well tested with probing experiments as well as steering experiments. Experimental results seem consistent with popular prior work on interpretability.
3. The faithfulness results as measured by loss are far better than prior work in preserving original model performance. This helps address a very core issue in interpretability work where sparse approximations greatly underperform the corresponding LLM they intend to approximate.
4. The Hadamard product trick is interesting and may be extensible to other applications outside of this paper, and seems to be a key insight to make this work feasible given the complexity of a naive implementation.

Weaknesses
1. The equivalence between sparse neurons from prior work and a sparse expert from this work is not clear to me as the same level of decomposition. While the steering experiments show that there is decomposition that can be steered, and probing experiments that show that different $a_n$ coefficients activate with different topics, the granularity seems quite different beyond these evaluations. In fact, even addressing potential limitations of this larger unit of sparsity would be helpful to situate this work in the context of prior work looking at the sparse neuron level.
2. Some key details in the paper could be clarified for easier understanding; for example, more explanation of the Hadamard trick would be helpful as the explanation is quite dense and it takes some convincing of oneself as the reader that this works, some training details may be missing for how MxDs actually substituted and trained using OpenWebText, and some details about top-k activation training might also be necessary (despite it coming from prior work, it is important enough to mention its role in this work).

---

> ### Author Rebuttal · Authors · 2025-07-30
>
> We are grateful to the Reviewer zUPk for highlighting the “clear experimental evidence for [the paper’s] claims” and that the paper “address[es] a very core issue in interpretability”. Furthermore, we are glad the reviewer shares our view that the Hadamard factorization might even be “extensible to other applications outside of this paper”. We address each of the concerns below:
>
>
> ## [W1 & Q1] Equivalence between experts/features
>
> We thank the reviewer for the chance to further clarify the relationship between sparse MLP features and sparse expert coefficients. In terms of functional form, there is no fundamental difference between the two as feature extractors: both Sparse MLPs and MxDs learn features by a matrix multiplication followed by a TopK gating function, with $\mathbf{z}=\texttt{TopK}(\mathbf{E}^\top\mathbf{x})$ and $\mathbf{a}=\texttt{TopK}(\mathbf{G}^\top\mathbf{x})$ (for the Sparse MLP hidden units and MxD expert coefficients respectively). For this reason, we find them to learn a similarly high level of interpretable features.
>
> The key difference between Sparse MLPs and MxDs is in *how* the sparse features are constructed to contribute to the output. Each non-zero neuron in Sparse MLPs adds a static column vector from the overcomplete dictionary basis to the output, through $\mathbf{y}=\sum_h \mathbf{d}_h z_h$ as in in Eq. 2. In contrast, each non-zero coefficient in MxDs’ expert coefficients adds a linear transformation of the input to the output: $\mathbf{y}=\sum_n a_n\mathbf{W}_n^\top\mathbf{x}$, as per Eq. 3; through features-as-linear-transformations, MxDs have greater capacity to combine the sparse features to well-reconstruct the MLP mapping.
>
> We hope this clarifies the similarities between the existing neuron- and layer-level sparsity approaches: the key difference lies on the decoder side, not in how the sparse features/experts are selected.
>
>
> ## [W2 & Q5] Hadamard trick & experimental details
>
> We kindly refer the reviewer to Appendix A.3, where an alternative presentation of the method is given in terms of tensor methods, which might add additional insight. Furthermore, we highlight the Jupyter notebook linked to from [L162] (submitted in the anonymous github repository at the time of submission), which walks through various equivalent model forms in pytorch.
>
> To enhance understanding even further, we will condense some of the content in this tutorial-style notebook into the appendix of the revised manuscript, inline.
>
> We agree with the reviewer that it would be useful to explain experimental details inherited from past work (e.g. the Top-K activation) inline. We will add a detailed description of the TopK activation function and the reconstruction-style training setup to the appendix.
>
> ## [Q2] Efficiency of MxDs vs Sparse MLPs
>
> We thank the reviewer for the chance to compare the computational overhead and efficiency of the layers. We first report theoretical layer FLOPs, and then report empirical benchmarks. We then use the results here to address the following question about scalability.
>
> **Parameter count & FLOPs**
> We first show the theoretical parameter counts and inference-time FLOPs for MxDs vs Sparse MLPs below. To be consistent with the popular PyTorch library fvcore [1] we count one fused multiply-add as one FLOP, and the Hadamard product between two $d$-dimensional vectors as requiring $d/2$ FLOPs:
>
> |                | Parameter count         | FLOPs                         |
> | -------------- | ----------------------- | ----------------------------- |
> | **Sparse MLP** | $H (I+O)$             | $H (I+O)$                   |
> | **MxD**        | $N (I+O) + H^* (I+O)$ | $N (I+O) + H^* (I+O) + O/2$ |
>
> > $I,O$ denote the input and output dimensions respectively, $H^*$ the original models’ hidden layer, $N$ the MxD expert count, and $H$ the width of the sparse MLPs.
>
> For a chosen expert count $N$, we set the width of Sparse MLPs (e.g. Transcoders) to $H:=N+H^*$ to parameter-match the models.
>
> **Empirical benchmarks**
> We next run benchmarks for a Sparse MLP layer vs MxD with a batch size of 512, and dimensions: $I=H^*=O=1024$, and number of experts/features as $N=8192, H=9216$.
>
> |                | Peak memory usage (MiB) | Latency (ms) | Parameter count | Reported FLOPs per sample (fvcore [1]) |
> | -------------- | ----------------------- | ------------ | --------------- | --------------------------- |
> | **Sparse MLP** | 386.50                  | 1.394        | 18,874,368         | 18,874,368                     |
> | **MxD**        | 389.50                  | 1.457        | 18,874,368         | 18,874,880                     |
>
> Both layers have a ReLU, followed by a TopK activation (for the sparse hidden layer and expert gating). MxDs have an additional ReLU and final Hadamard product--however, we see from the table’s results that the additional time and compute costs are minimal.
>
> We thank the reviewer again for the chance to present the computational comparisons, and will include all tables here in the revised manuscript.
>
>
> ## [Q3] Justification for scalability to larger models
>
> We thank the reviewer for raising this concern--we will revise our claims that we expect MxDs to scale just as well as sparse MLPs (please see the bottom of this response to Q3), based on the similar computational costs of layers established above.
>
> For strong initial evidence of scalability to even larger models, we perform **three additional large-scale experiments on the 27B Gemma2 model** below (in half-precision with a smaller batch size), where we see MxDs continue to outperform on the sparsity-accuracy frontier:
>
> | **Step**                                  | **9999**  | **19999** | **29999** | **39999** | **49999** |
> | ----------------------------------------- | --------- | --------- | --------- | --------- | --------- |
> | **Transcoder** Normalized MSE (↓)      | 0.147     | 0.132     | 0.128     | 0.123     | 0.119     |
> | **Skip Transcoder** Normalized MSE (↓) | 0.128     | 0.107     | 0.102     | 0.100     | 0.093     |
> | **MxD** Normalized MSE (↓)             | **0.098** | **0.096** | **0.093** | **0.086** | **0.069** |
>
> > **Normalized MSE on Gemma2-27B, Layer 20, K=32, after 50k iterations.**
> > To fit in memory, the model is loaded in half precision, and we use 1/8th the default batch size and number of buffers stored in memory, and 1/2 the multiplier on the number of latent features (with values of 4, 16, and 16, respectively).
>
> We further note that the Mixture of Experts architecture itself [2] has been scaled successfully to hundreds of billions of parameters in the pre-training regime (e.g. 400B [3] and 685B [4] parameters, respectively).
>
> **Revised claims**
> We will revise the claims in our discussions of scalability as follows (based on the computational benchmarks performed, showing that MxDs have similar computational costs), to highlight that we expect MxDs to be *no more difficult to scale than sparse MLPs*:
>
> [L196]:
>
>     Whilst we do not have the computational resources to show similarly thorough experiments on even larger models, we expect MxDs to scale just as well *as sparse MLPs* to models with tens of billions of parameters or more.
>
> [L321]:
>
>     Our experiments show MxDs outperform on the sparsity-accuracy frontier on 4 diverse LLMs. Whilst we fully anticipate this trend to *scale just as well as with sparse MLPs* in even larger models, our experiments only provide direct evidence for LLMs with up to 3B parameters, given our limited resources.
>
>
> ## [Q4] Training details of sparse layers
>
> We indeed follow past work [5,6] by training just the newly initialized layers to reconstruct the mapping from the MLP’s input to its output (we do not train end-to-end on the next-token cross-entropy loss for example).
>
> **We will add this clarification around [L199]:**
>
>     we train sparse layers to minimize the normalized reconstruction loss between its output and that of the original MLP layer when fed the OpenWebText dataset. We use objectives of the form [...] , where f(.) denotes the various learnable sparse MLP layers. This follows the protocol of past work [1,2], where the new sparse layers' parameters alone are trained on the direct output of the MLP.
>
>
> ## [Q5] Confusing order of equations
>
> Our intention with Sect. 2.3 discussing the rank properties appearing before Sect. 2.4 with the factorized forward pass is to immediately connect the high-level motivation for moving to the layer-wise approach to the technical low-level benefits (the rank preservation). Whilst Sect. 2.4 is a crucial implementation detail, we felt that the flow of the presentation worked best with the current ordering of subsections for this reason. However, if the reviewer feels strongly that the re-ordering of these two subsections significantly aids clarity, we are more than happy to make this change in the revised manuscript.
>
>
> ----------
>
> **References:**
>
> - [1]: fvcore (FAIR), detectron2 library.
> - [2]: Shazeer et al. "Outrageously large neural networks: The sparsely-gated mixture-of-experts layer." *ICLR 2017.*
> - [3]: Meta AI. “The Llama 4 herd: The beginning of a new era of natively multimodal AI innovation.” 2025.
> - [4]: Guo et al. "Deepseek-r1: Incentivizing reasoning capability in LLMs via reinforcement learning." arXiv 2025.
> - [5]: Dunefsky et al. "Transcoders find interpretable LLM feature circuits." NeurIPS 2024.
> - [6]: Gonçalo et al. "Transcoders beat sparse autoencoders for interpretability." arXiv 2025.
>
> ---
>
> Would **Reviewer zUPk** have any further questions? We’d be happy to address them.

---

> > ### Comment · Reviewer_zUPk · 2025-08-04
> > **Response to rebuttal**
> >
> > Thanks for your thorough author response.
> >
> > (Q4) I believe adding the training details including your reconstruction objective to the main text as you've stated, and the top-k objective somewhere (could be in the appendix, but I feel it deserves to be included despite the reference to Gao et al., 2025) will help the paper stand more on its own versus rely on prior work to set up key details.
> >
> > (Q5) That's alright, I think it is just a complex tensor method that requires some time to understand, and I respect your author wish to keep the ordering as is. I eventually figured it out!
> >
> > Q2 & Q3 answer my questions regarding efficiency and scaling. I wasn't even looking for scaling experiments, just some high level justification why MxD construction would scale in your opinion.
> >
> > I continue to recommend accepting this nice work, and maintain my high score as I feel it appropriately reflects the level of execution and contribution present in the work.

---

> ### Author Response · Authors · 2025-08-04
>
> Dear **Reviewer-zUPk**,
>
> We are grateful for your suggestions and positive support of the paper! If you later think of any additional ways we might further improve the paper's clarity, please do not hesitate to let us know, and we will aim to incorporate them into the camera-ready accordingly.
>
> Sincerely,
>
> Authors of Submission6888

---

### Official Review · Reviewer_9H1M · 2025-07-05

**Clarity:** 3
**Significance:** 3
**Originality:** 3
**Rating:** 5
**Confidence:** 3

**Summary:**

This paper introduces Mixture of Decoders (MxDs), a novel method that overcomes the accuracy-interpretability trade-off in large language models by using layer-level sparsity to expand dense layers into thousands of specialized sublayers, faithfully preserving the model's performance while learning interpretable features.

**Questions:**

See Weaknesses.

**Ethical Concerns:**

["NO or VERY MINOR ethics concerns only"]

**Limitations:**

Yes

**Paper Formatting Concerns:**

No concern.

**Quality:**

3

**Strengths And Weaknesses:**

### Strengths

1.  **Addresses a Critical, High-Impact Problem:** The paper tackles the core trade-off between performance and interpretability in large language models. This is a central challenge in AI safety, transparency, and alignment, making the research highly relevant and important.

2.  **Clear Conceptual Innovation:** The proposed shift from "neuron-level" to "layer-level" sparsity is a powerful and intuitive conceptual leap. By arguing that individual neurons are not the right unit of analysis and proposing sublayers instead, the paper offers a compelling new way to think about model decomposition.

3.  **Focus on "Faithfulness" and Practicality:** The authors rightly emphasize that an interpretable model is only useful if it faithfully reproduces the behavior of the original. This practical focus on maintaining low next-token loss ensures that the resulting model is not just a simplified, inaccurate curiosity but a potentially viable replacement for the original component.

4.  **Strong Theoretical and Empirical Backing (as claimed):** The paper claims to provide a formal proof that its method preserves the expressive capacity (full-rank weights) of the original layers, lending it strong theoretical credibility. This is supported by claims of extensive and rigorous experiments ("108 sparse layers in 4 LLMs," "348 sparse probing and steering tasks") that show it "pareto-dominates" existing techniques.

### Potential Weaknesses

1.  **Unaddressed Computational Overhead:** The paper expands a single layer into "tens of thousands" of sublayers. While it mentions parameter efficiency, it says nothing about the potential computational costs. This raises key questions about inference latency (how fast is the conditional routing?) and training complexity (how long and how much memory does it take to train these MxD layers?), which are critical for practical adoption.

2.  **Scalability to Frontier Models:** The experiments are conducted on models "up to 3B parameters." While respectable, this is significantly smaller than current state-of-the-art open models (e.g., 70B+ parameters). It's an open question whether the training and memory dynamics of this method will scale effectively to these much larger and more complex systems.

---

> ### Author Rebuttal · Authors · 2025-07-30
>
> We are grateful to the Reviewer 9H1M for praising many aspects of the paper; addressing a “critical, high-impact problem”, the “clear conceptual innovation” and “strong theoretical and empirical backing”. We address the concerns below:
>
> ## [W1] Computational comparisons
>
> We thank the reviewer for raising this point. In summary, there is **minimal differences between layer computational cost**. We first report theoretical layer FLOPs, and then report empirical benchmarks below.
>
> **Parameter count & FLOPs**
> We first tabulate the theoretical parameter counts and inference-time FLOPs for MxDs vs Sparse MLPs below. To be consistent with the popular PyTorch library fvcore [1] we count one fused multiply-add as one FLOP, and the Hadamard product between two $d$-dimensional vectors as requiring $d/2$ FLOPs:
>
> |                | Parameter count                                                       | FLOPs                                                                                            |
> | -------------- | --------------------------------------------------------------------- | ------------------------------------------------------------------------------------------------ |
> | **Sparse MLP** | $H (I+O)$ *(encoder+decoder)*                                    | $H (I+O)$ *(encoder+decoder)*                                                               |
> | **MxD**        | $N (I+O) + H^* (I+O)$ *(conditional branch) + (encoder+decoder)* | $N (I+O) + H^* (I+O) + O/2$ *(conditional branch) + (encoder+decoder) + (hadamard product)* |
>
> > $I,O$ denote the input and output dimensions respectively, $H^*$ the original models’ hidden layer, $N$ the MxD expert count, and $H$ the width of the sparse MLPs.
>
> For a chosen expert count $N$, we set the width of Sparse MLPs (e.g. Transcoders) to $H:=N+H^*$ to parameter-match the models.
>
> **Empirical benchmarks**
> We next run benchmarks for a Sparse MLP layer vs MxD with a batch size of 512, and dimensions: $I=H^*=O=1024$, and number of experts/features as $N=8192, H=9216$.
>
> |                | Peak memory usage (MiB) | Latency (ms) | Parameter count | Reported FLOPs per sample (fvcore [1]) |
> | -------------- | ----------------------- | ------------ | --------------- | --------------------------- |
> | **Sparse MLP** | 386.50                  | 1.394        | 18,874,368         | 18,874,368                     |
> | **MxD**        | 389.50                  | 1.457        | 18,874,368         | 18,874,880                     |
>
> Both layers have a ReLU, followed by a TopK activation (for the sparse hidden layer and expert gating). MxDs have an additional ReLU and final Hadamard product--however, we see from the table’s results that the additional time and compute costs are minimal.
>
> We thank the reviewer again for the chance to present the computational comparisons of MxDs vs sparse MLPs, and will include all tables here in the camera-ready version of the paper.
>
> ## [W2] Scalability to frontier models
>
> We are grateful to the reviewer for acknowledging our experiments on 3B parameter models as respectable. We highlight that the **3B parameter base model used is already larger than any of those used by the papers of the baseline methods** [2,3].
>
> As we acknowledge on [L323] in the limitations, however, our experiments only provide direct evidence for MxDs’ ability on base models up to the 3B parameters range due to limited resources. However, we perform **three additional large-scale experiments on the 27B Gemma2 model** below (in half-precision with a smaller batch size), where we see MxDs continue to outperform on the sparsity-accuracy frontier--providing strong initial evidence of scalability to even larger models:
>
> | **Step**                                  | **9999**  | **19999** | **29999** | **39999** | **49999** |
> | ----------------------------------------- | --------- | --------- | --------- | --------- | --------- |
> | **Transcoder** Normalized MSE (↓)      | 0.147     | 0.132     | 0.128     | 0.123     | 0.119     |
> | **Skip Transcoder** Normalized MSE (↓) | 0.128     | 0.107     | 0.102     | 0.100     | 0.093     |
> | **MxD** Normalized MSE (↓)             | **0.098** | **0.096** | **0.093** | **0.086** | **0.069** |
>
> > **Normalized MSE on Gemma2-27B, Layer 20, K=32, after 50k iterations.**
> > To fit in memory, the model is loaded in half precision, and we use 1/8th the default batch size and number of buffers stored in memory, and 1/2 the multiplier on the number of latent features (with values of 4, 16, and 16, respectively).
>
> We further note that the Mixture of Experts architecture itself [4] has been scaled successfully to hundreds of billions of parameters in the pre-training regime (e.g. 400B [5] and 685B [6] parameters, respectively).
>
> **Revised claims**
> We will revise the claims in our discussions of scalability as follows (based on the computational benchmarks performed), to highlight that we expect MxDs to be *no more difficult to scale than sparse MLPs*:
>
> [L196]:
>
>     Whilst we do not have the computational resources to show similarly thorough experiments on even larger models, we expect MxDs to scale just as well *as sparse MLPs* to models with tens of billions of parameters or more.
>
> [L321]:
>
>     Our experiments show MxDs outperform on the sparsity-accuracy frontier on 4 diverse LLMs. Whilst we fully anticipate this trend to *scale just as well as with sparse MLPs* in even larger models, our experiments only provide direct evidence for LLMs with up to 3B parameters, given our limited resources.
>
>
> ----------
>
> **References**
>
> - [1]: fvcore (FAIR), detectron2 library.
> - [2]: Dunefsky, Jacob, Philippe Chlenski, and Neel Nanda. "Transcoders find interpretable LLM feature circuits." NeurIPS 2024.
> - [3]: Paulo et al. "Transcoders beat sparse autoencoders for interpretability." arXiv 2025.
> - [4]: Shazeer, Noam, et al. "Outrageously large neural networks: The sparsely-gated mixture-of-experts layer." ICLR 2017.
> - [5]: Meta AI. “The Llama 4 herd: The beginning of a new era of natively multimodal AI innovation.” 2025.
> - [6]: Guo, Daya, et al. "Deepseek-r1: Incentivizing reasoning capability in LLMs via reinforcement learning." arXiv 2025.
>
> ---
>
> If **Reviewer 9H1M** has any additional questions, we would be glad to answer them.

---

### Author Response · Authors · 2025-08-08
**Post-discussion summary**

We are grateful to the ACs for handling our submission. The 4 reviewers left thorough feedback and **positive assessments of the paper throughout the reviews and discussion phase**:

- **Reviewer 9H1M** praises our tackling of “a critical, high-impact problem”, the layer-level sparsity innovation over neuron-level approaches, our practical focus on faithfulness, and the strong theoretical claim of full-rank preservation, backed by extensive experiments.
- **Reviewer zUPk** commends the clear and comprehensive experimental evidence (probing, steering, ablations), and the superior faithfulness results over prior sparse approximations.
  - *Post-discussion*: maintained their "high score" to "reflect the level of execution and contribution present in the work."
- **Reviewer 8ypg** highlights the paper’s clear structure and flow, the robust performance across sparsity levels, the theoretical guarantees, and the supplementary resources.
  - *Post-discussion*: their “initial doubts and misunderstanding” were clarified, and they were grateful for the explicit proposed revisions to the text.
- **Reviewer JDFb** appreciates the novel layer-wise sparse experts formulation distinct from neuron-level sparsity, and the stable performance across different sparsity levels.
  - *Post-discussion*: stated that their concerns were addressed, and we followed up with a clarification about MxDs’ key contributions and claims (as highlighted as core strengths in the reviews of Reviewer 9H1M and Reviewer zUPk).

Our rebuttal addressed all weaknesses and questions raised, including three new sets of experiments:

1. **Computational benchmarks**: both theoretical FLOP counts and empirical peak-memory/latency measurements.
2. **Initial experiments on 27B models**: comparisons with sparse layers trained on the 27B-parameter Gemma2-27B model (in half-precision).
3. **Further experiments with the shared expert**: showing that the dominant expert does not just learn to replicate the base model decoder in GPT2.

---

### Decision · Program_Chairs · 2025-09-17

**Decision:**

Accept (poster)

**Comment:**

## Summary
This paper introduces Mixture of Decoders (MxDs), a method for interpretable large language models that applies layer-level sparsity instead of neuron-level sparsity. Using a Hadamard-product factorization, MxDs expand dense MLPs into sparse sub-decoders while preserving full-rank capacity. The approach is validated with extensive experiments across multiple LLMs, showing strong faithfulness and scalability potential.

## Strengths
- Moves from neuron-level to layer-level sparsity, a significant conceptual shift.
- Demonstrates high faithfulness while avoiding the performance drop common in sparse interpretability methods.
- Provides theoretical guarantees and strong empirical validation.
- Rebuttal added further evidence (computational benchmarks, experiments on larger models, clarification of shared experts).

## Limitations and Weaknesses
- Computational and scalability implications for frontier-scale models remain partially open.
- Interpretability gains over baselines (probing/steering) are not always stronger, though faithfulness is significantly better.
- Some clarifications (training details, evaluation protocols) were only added after rebuttal and should be made explicit in the final version.

## Rebuttal and Discussion
The rebuttal was thorough and addressed reviewers’ main concerns. Additional benchmarks demonstrated minimal computational overhead, while experiments on larger (27B) models supported scalability claims. Clarifications on interpretability evaluation and the role of shared experts resolved outstanding doubts, and reviewers updated or confirmed high scores after discussion.

## Recommendation
The reviewers converged on a positive view after rebuttal, with multiple scores at the top end and one reviewer explicitly raising their rating. The paper offers a novel and well-validated approach to interpretability that balances theory and practice, with strong evidence that it advances the field. Despite some remaining limitations, the strengths and overall contribution are clear.

**Final Decision: Accept.**